# Challenges in Matrix Metalloproteinases Inhibition

**DOI:** 10.3390/biom10050717

**Published:** 2020-05-05

**Authors:** Helena Laronha, Inês Carpinteiro, Jaime Portugal, Ana Azul, Mário Polido, Krasimira T. Petrova, Madalena Salema-Oom, Jorge Caldeira

**Affiliations:** 1Centro de Investigação Interdisciplinar Egas Moniz, Instituto Universitário Egas Moniz, 2829-511 Caparica, Portugal; h.laronha@campus.fct.unl.pt (H.L.); icarpinteiro@egasmoniz.edu.pt (I.C.); aazul@egasmoniz.edu.pt (A.A.); mpolido@egasmoniz.edu.pt (M.P.); moom@egasmoniz.edu.pt (M.S.-O.); 2UCIBIO and LAQV, Requimte, Faculdade de Ciências e Tecnologia, Universidade Nova de Lisboa, 2829-516 Caparica, Portugal; k.petrova@fct.unl.pt; 3Faculdade de Medicina Dentária Universidade de Lisboa, 1649-003 Lisboa, Portugal; jaime.portugal@fmd.ulisboa.pt

**Keywords:** matrix metalloproteinases, TIMP, synthetic inhibitors

## Abstract

Matrix metalloproteinases are enzymes that degrade the extracellular matrix. They have different substrates but similar structural organization. Matrix metalloproteinases are involved in many physiological and pathological processes and there is a need to develop inhibitors for these enzymes in order to modulate the degradation of the extracellular matrix (ECM). There exist two classes of inhibitors: endogenous and synthetics. The development of synthetic inhibitors remains a great challenge due to the low selectivity and specificity, side effects in clinical trials, and instability. An extensive review of currently reported synthetic inhibitors and description of their properties is presented.

## 1. Introduction

Matrix metalloproteinases (MMPs) are a protein family within the metzincin superfamily, comprising zinc-dependent endopeptidases with similar structural characteristics but with different substrate preferences. MMPs are produced and secreted from cells as inactive proenzymes depending, herein, on a structural alteration for activation [1,2,3,4,5,6]. In human tissues, there are 23 different types of MMPs expressed and they can be subdivided according to their substrate specificity, sequential similarity, and domain organization [1,2,4,7,8,9,10,11,12,13,14,15,16,17] (Table 1). 

The most common structural features shared by MMPs are [1,2,4,5,7,8,10,11,12,13,14,16,18] (Figure 1) a pro-domain, a catalytic domain, a hemopexin-like domain, and a transmembrane domain for membrane type MMPs (MT-MMPs) although some MMPS do not have all the structural features represented in the figure. The pro-domain keeps MMP inactive by a cysteine switch, which interacts with the catalytic zinc making it impossible to connect the substrate. The catalytic domain has two zinc ions, three calcium ions, and three histidine residues, which are highly conserved [1,2,3,4,5,6,7,8,9,11,12,13,14,15,16,17,18,19,20]. In the terminal zone of the catalytic domain there is a region that forms the outer wall of the S_1_’ pocket [1,14,17]. This pocket is the most variable region in MMPs and it is a determining factor for substrate specificity [1,2,6,7,11,17,18]. However, there are six pockets (P_1_, P_2_, P_3_, P_1_’, P_2_’, and P_3_’) and the fragments of the substrates or inhibitors are named depending on the interaction with these pockets (R_1_, R_2_, R_3_, R_1_’ or R_a_, R_2_’, and R_3_’). The linker is proline-rich, of variable length, allowing inter-domain flexibility and enzyme stability [4,8,12,13]. The hemopexin-like domain is necessary for collagen triple helix degradation and is important for substrate specificity [3,4,7,9,19].

The MMPs can process ECM proteins and glycoproteins, membrane receptors, cytokines, hormones, chemokines, adhesion molecules, and growth factors [1,3,4,6,7,9,10,11,13,14,20,21,22,23,24,25,26]. However, the presence and the activity of MMPs have been demonstrated to be intracellular [25,26]. For example, some studies show intracellular localization of MMP-2 in cardiac myocytes and colocalization of MMP-2 with troponin I in cardiac myofilaments [23]. The MMP-2 activity has also been detected in nuclear extracts from human heart and rat liver [23]. The MMPs are involved in many biologic processes, such as tissue repair and remodulation, cellular differentiation, embryogenesis, angiogenesis, cell mobility, morphogenesis, wound healing, inflammatory response, apoptosis, ovulation, and endometrial proliferation [1,2,4,6,8,10,11,13,16,17,18,20,27]. The deregulation of MMPs activity leads to the progression of various pathologies depending on which enzyme is involved [1,6,10,13,14,15,16,17,20,27]: cancer and metastasis, inflammatory processes, arthritis, ulcers, periodontal diseases, brain degenerative diseases, liver cirrhosis, fibrotic lung diseases, otosclerosis, atherosclerosis, multiple sclerosis, dilated cardiomyopathy, aortic aneurysm, or varicose veins.

Although therapeutic strategies for specific inhibition of MMPs have been long researched, they are difficult to develop because these enzymes are involved in a myriad of pathways [2,5]. However, this inhibition can be done at the biomolecular expression and active enzyme terms [2,5,18]. The MMPs inhibitors can be divided into endogenous inhibitors, which can be specific or non-specific, and synthetic inhibitors [1,2,4,7,10,12,13,14,16,20,28,29] (Table 2).

## 2. Specific Endogenous Inhibitor-Tissue Inhibitors of Metalloproteinases (TIMPs)

Tissue inhibitors of metalloproteinases (TIMPs) are endogenous proteins responsible for the regulation of MMPs activity, but also of families such as the disintegrin metalloproteinases (ADAM and with thrombospondin motifs ADAMTS) and therefore for maintaining the physiological balance between ECM degradation and MMPs activity [1,2,8,9,18,30]. There are four TIMPs (TIMP-1, -2, -3, and -4) (Table 3), with 22–29 KDa and 41%–52% sequential similarity [2,4,12,13,16,20,31].

TIMPs consist of a *N*- and *C*-terminal domain with 125 and 65 amino acids, respectively, each containing six conserved cysteine residues, which form three conserved disulphide bonds [2,4,7,8,9,12,31,32] (Figure 2a). The *N*-terminal domain is an independent unit, which can be inhibited by MMPs, in a 1:1 ratio [2,4,8,9,10,12,13,16,20]. This domain has two groups of four residues: Cys-Thr-Cys-Val and Glu-Ser-Val-Cys (Figure 2b), which are connected by disulphide bounds which are important for TIMP activity [7,12]. This is the main domain responsible for MMP inhibition through its binding to the catalytic site in a substrate-like manner [31]. The several domains allow the TIMP and pro-gelatinases interactions [4].

## 3. Non-Specific Endogenous Inhibitors

Non-specific endogenous inhibitors have been reported to inhibit MMPs (Table 4), however, the inhibition mechanism details have only been partially discovered [7,12].

Human α2-macroglobulin is a glycoprotein with four identical subunits that act by entrapping MMP and the complex is cleared by endocytosis [2]. The α2-macroglobulin has been found in blood and tissue fluid [2,31]. The tissue factor pathway inhibitor (TFPI) is a serine proteinase inhibitor, which targets MMP-1 and -2, but this inhibition mode is still unknown [7,12]. The *C*-terminal proteinase enhancer protein and tissue factor pathway inhibitor have sequences with certain similarities to the *N*-terminal domain of TIMPs [31].

## 4. Synthetic Inhibitors

MMPs are molecular targets for the development of therapeutic and diagnostic agents [14]. The development of synthetic MMP inhibitors was initially based on the peptide sequence, recognized by proteases, with different chemical functionalities, capable of interacting potently with zinc ion [11,12,13,19]. The requirements for an effective inhibitor are [2,11,13,17,19,35]:-A functional group able to chelate the zinc ion (II)-zinc binding group (ZBG). The first generation inhibitors used hydroxamate (CONHO^−^) but the second generation use carboxylate (COO^−^), thiolates (S^−^), phosphonyls (PO_2_^−^), for example (Figure 3);-At least one functional group that promotes hydrogen bonding with the protein backbone;-One or more side chains undergoing Van der Waals interactions with enzyme subsites.

ZBGs have negative charges that prevent their penetration in the cell, restricting their activity to the extracellular space which reduces their cell toxicity [2]. Changes in the ZBG structure or in the point of attachment of the ZBG to the backbone of the MMP inhibitor can change its potency and selectivity [2]. When comparing what selectivity or potency of different ZBGs leaving the structure constant, Castelhano et al. arrived at the following list [36]: hydroxamic acid >> formylhydroxylamine > sulfhydryl > aminocarboxylates > carboxylate. 

The selectivity of inhibitor is a primordial goal of MMPs’ inhibitors (MMPis) design to increase efficacy and prevent side effects [2]. This selectivity is based on two molecular characteristics [11]: a chelating capacity for catalytic zinc and the presence of hydrophobic bridges of the active center for the S_1_’ pocket. Numerous strategies have been suggested for creating selective MMP inhibitors [27] (Figure 4): endogenous-like inhibitors, exosite targeting inhibitors, a combination of exosite binding and metal chelating inhibitors, and function-blocking antibodies.

The synthetic inhibitors have had a great challenge in their development since first it is necessary to identify the enzymes that are involved in disease progression. Moreover, this goal has an additional difficulty as there are more than 50 human metalloproteinases (23 MMPs, 13 ADAM, and 19 ADAMTS) [2].

### 4.1. Hydroxamate-Based Inhibitors

The first generation of MMPis (1995–1999 [16]) were designed based on the knowledge of the triple helix collagen amino acid sequence (cleavage site) and the information derived from specific substrates [6,15,19,31,35,37]. These compounds contain a hydroxamic acid group as ZBG [5,6,15,18,27,28,31,37]. Hydroxamic acids (HA) were first described in 1986 [38,39]. They are easy to synthesize, are monoanionic compounds, bidentate chelating agents, and due the excellent zinc-chelating capability they are the more popular ZBG for MMPs [2,6,17,18,27,28,29]. The hydroxamic group established interactions with zinc ions, through two oxygens and two hydrogen bonds (NH and OH groups of HA with Ala and Glu, respectively), forming a distorted triangular bipyramid [2,5,18,19,28,29] (Figure 5a). Reich et al., in 1988 [40], used a hydroxamic acid compound, S*C*-44463 (Figure 5b), to block collagenase and prevent metastasis in a mouse model, which initiated the era of MMP inhibition therapeutics [40].

The structure–activity relationship (SAR) studies for a series of hydroxamic acids with a quaternary carbonyl group at R_1_ suggested that [13] (Figure 6):(i)The stoichiometric orientation of the substituent at R1 position is crucial for the activity;(ii)The phenylpropyl group was established as the best substituent at position R1;(iii)Hydrophobic substituents at R_2_’ position and *N*-metiamides at R_3_’ position were considered as the most appropriate.

Modification in Ra position (α substituent): a beneficial effect is conferred by lipophilic substituents capable of hydrogen bonding [19]. Analogues of Marimastat have been reported, where the α position is disubstituted by a hydroxyl group and a methyl group [19] (Figure 7a). The X-ray structures of the MMP-3-inhibitor complex showed that there were hydrogen bonds between the hydroxyl group and Ala_165_ [19]. By binding the positions Ra and R_2_ in a single chain, forming a cyclic inhibitor (Figure 7b), there was a substantial increase in water solubility [19,29,41,42]. This strategy led to the discovery of two inhibitors with similar potency to non-cyclized analogues, SE205 and SC903 [41] (Figure 7c,d). 

The introduction of conformational restrictions through the addition of a three-membered ring between positions α and R_1_ (Figure 8a) has been reported by Martin et al. [43], and it resulted in reduced inhibition of MMP-9 [43]. However, the introduction of a six-membered ring between these same positions resulted in the inactivity of the compound [19] (Figure 8b).

Exploring the depth of S_1_’ pocket, bulky groups in Rα position confer selective inhibition for MMP-2, -8, and -9 [29]. An example is the presence of a biphenyl group, which showed higher inhibitory activity against MMP-9 [29].

Modification in R_1_ position: the incorporation of long groups in the R_1_ position can promote the selectivity of MMPis, since pocket S_1_’ can undergo conformational changes to accommodate certain substituents [19].

Broadhurst et al. showed that an alkyl chain (C_9_) at the R_1_ position reduces the inhibition of MMP-1, but maintains the inhibitory activity against MMP-2, -3, and -9 [19] (Figure 9a). For matlystatin derivates in the C_9_ chain, R-94138 (Figure 9b) promotes the inhibition against MMP-9 [44]. The succinyl hydroxamates analogues in the C_9_ chain promote selectivity for MMP-2, however, a C_10_ chain (Figure 9c) results in MMP-1 inhibition, while a further increase of the chain to C_16_ (Figure 9d) leads to a loss of activity against MMP-1 [19].

Replacement of the R_1_-R_2_ bond of succinyl hydroxamates acid inhibitors by a sulfonamide bond (Figure 10a) results in substantial loss of inhibitory activity because the hydrogen bond (C=ONH; Figure 10b) with leucine is stronger than the new sulfonyl oxygen bond, due to the pyramidal nature of the sulfonamide [19].

Modifications in R_2_ position: modifications in the R_2_ position led to a modest effect in inhibitory activity, in vitro, and affects the pharmacokinetic properties [29]. Marimastat and Ro31-9790 (Figure 11) have a good oral activity because the bulky tert-butyl group assists the adjacent amide bond during absorption from an aqueous environment to the lipid environment of the cell membrane [45]. The beneficial combination of the tert-butyl group with the α-hydroxyl group increases the water solubility [19,29]. Babine and Bender suggest that the tert-butyl as R_2_ group leads to less Van der Waals interactions, comparing with other groups [46].

Ikeda et al. described compounds with phenyl R_2_ substituents (Figure 12) (KB-R7785), which are active orally, due to the beneficial effect of the R_2_ phenyl group on absorption, where the amide shielding and lipophilicity may assist in transepithelial resorption [47]. This inhibitor shows activity against MMP-1 in rats and its effectiveness in arthritis has been demonstrated [47].

Modifications in R_3_ position: the S_3_’ pocket is an open area and several groups can be introduced at the R_3_ position [19]. The introduction of the benzhydryl group leads to compounds with selectivity to the MMPs-3 and -7 [19].

#### 4.1.1. Succinyl Hydroxamic Acid-Based Inhibitors

Succinyl hydroxamate derivates can be subdivided to peptide derivatives or non-peptide compounds [48]. The *N*-acetylcysteine has been reported to affect the tumoral invasion process and metastasis by MMP-2 and -9 inhibition [49]. The L-cysteine-2-phenylethylamide is an effective inhibitor, in which the phenyl group fills the S_1_’ pocket of MMP-8 [50]. Foley et al. prepared several dipeptides derivatives containing cysteine (**R**CO-Cys-**AA**-NH_2_) and concluded [51]:-The variation of the acyl group and the second amino acid (**AA**) leads to the activity against different MMPs.-The **R** group interacts with the S_1_’ pocket.

Batimastat (Table 5) was the first MMPi to enter in clinical trials for cancer as it inhibits MMP-1, -2, -7, and -9, but, due to its poor oral bioavailability, it was superseded by Marimastat [28,29,31] (Table 5), which has an alpha-hydroxyl group increasing the aqueous solubility [29]. Marimastat inhibits the activity of MMP-1, -2, -3, -7, -9, -12, and -13 [31]. However, Marimastat failed in clinical trials due to the absence of a therapeutic effect and the patients treated developed musculoskeletal toxicity (MST) [6,31]. Batimastat, marimastat, and ilomastat are examples of succinyl hydroxamates, which have very analogous structure to that of collagen and inhibit MMPs by bidentate chelation of the Zn^2+^ [2,6,29].

Several studies by Jonhson et al. [52] demonstrated that derivatives of succinyl hydroxamic acid (Figure 13a) are more potent for MMP-1 than the corresponding malonyl (Figure 13b) or glutaryl (Figure 13c) derivates. 

Marcq et al. [53] developed succinyl hydroxamates derivates selective for MMP-2, by modifications on Ilomastat structure (Figure 14a) to increase the overall hydrophobicity and, consequently, the selectivity [53]. This study resulted in a compound (Figure 14b) with an isobutylidene group of E geometry, which showed a 100-fold greater selectivity for MMP-2 over MMP-3, that is a 70-fold increase compared to Ilomastat [53].

Table 6 shows the IC_50_ and Ki values of some succinyl hydroxamic acid-based inhibitors [6,15,16,17,18,19,29,35,37,54,55].

#### 4.1.2. Sulfonamide Hydroxamic Acid-Based Inhibitors

In 1995, Novartis described the CGS-27023A (Figure 15a), a non-peptidic MMP-3 inhibitor, which has good oral availability but did not succeed in clinical trials [56]. The isopropyl group slows down the metabolization of the adjacent hydroxamic acid group and the 3-pyridyl substituent may aid partitioning into the hydrated negatively charged environment of the cartilage [56]. By analysis of the cocrystal structure of this inhibitor and MMP-12, it was possible to conclude that the binding mode between the hydroxamate moiety and the catalytic zinc ion was the same as the binding mode of hydroxamate-based inhibitors [6]. The interaction of CGS-27023A with MMP-3 was possible due to the p-methoxy phenyl substituent occupation of the S_1_’ pocket and the pyridylmethyl and isobutyl groups occupation of the S_2_’ and S_1_ pockets, respectively [29,56]. The modification of α to form the thioester derivate led to an increase of the inhibition of the deep pocket of the MMPs [29,56] (Figure 15b).

The NNGH (Figure 16a) (*N*-Isobutyl-*N*-(4-methoxyphenylsulfonyl)glycyl hydroxamic acid) was the starting point to many potent MMPis and is accommodated in the entry of the S_1_’ pocket, but does not penetrate it [6]. Barta et al. described a series of arylhydroxamate sulphonamides, active against MMP-2 and -13 (Figure 16b) [57]. In this compound, the sulfonyl group formed a single hydrogen bond with Leu_160_ and the piperidine-*O*-phenyl moiety extends into the S_1_’ pocket by Van der Waals interactions [57]. Noe et al. described a series of 3,3-dimethyl-5-hydroxy pipecolic hydroxamic acid, which possess potent inhibitory activity for MMP-13 [58]. In the first series of compounds, the 3-position of the piperidine ring was explored by the introduction of a polar functionality and it resulted in a compound with excellent activity on MMP-13 (Figure 16c), improved bioavailability, and lower metabolic clearance [58]. 

The incorporation of a cyclic quaternary center α led to a strong inhibitory effect against MMP-1, -2, -3, -8, -9, -12, and -13 [29]. The RS-113,456 (Figure 17a) is an inhibitor with better oral bioavailability and metabolic stability compared to the hydroxamate derivates [35]. These two features were improved by shifting the cyclic group to the α-position of the hydroxamic acid (Figure 17b) [29].

Table 7 shows the IC_50_ and Ki values of some sulfonamide hydroxamic acid-based inhibitors [6,15,16,17,18,19,29,35,37,54,55].

#### 4.1.3. Phosphamides Hydroxamic Acid-Based Inhibitors

The hydroxamic acids based on phosphamides are effective as MMPis due to the electronic environment of the phosphor atom [11]. The replacement of sulphonamide group by phosphinamide group leads to a potent inhibitor of MMP-3 (Figure 18), the collagenases and gelatinases [29]. The interactions between this inhibitor and the MMP-3 are realized by the phosphinamide phenyl group, that accommodates into the S_1_’ pocket and by the phosphinamide oxygen, which establishes the hydrogen bonds with NH of Leu_164_ and Ala_165_ [29]. However, this group is susceptible to hydrolysis at low pH, limiting the inhibitory activity [29]. 

All hydroxamate-based inhibitors are very potent and they inhibit MMPs at low concentrations [18]. On the other hand, the hydroxamate acids have poor oral bioavailability, inhibit multiple MMPs, and cause side effects [2,17,27,28,35]. Additionally, this functional group may be metabolized via dehydroxylation or may be cleaved by endopeptidases and releasing hydroxylamine, can be hydrolyzed to carboxylic acids or reduced to *O*-glucuronyl or *O*-sulfate, leading to decreased effective inhibitor concentration and reducing its potency in vivo [18,27,28,35].

Table 8 shows the IC_50_ and Ki values of some phosphamides hydroxamic acid-based inhibitor [6,15,16,17,18,19,29,35,37,54,55].

### 4.2. Non-Hydroxamate-Based Inhibitors

The side effects caused by hydroxamate-based inhibitors, due to the lack of selectivity and in vivo lability, have been fostering the development of new compounds with alternative ZBGs [5,6,11,16,17,27,28,29]. The second generation of MMPis (1999–2003 [16]) was designed with a wide variety of peptidomimetic and non-peptidomimetic structures with higher selectivity and exploiting the deep S_1_’ pocket present in some MMPs [6,15,16,18,59,60,61]. These compounds include carboxylic acids, sulfonylhydrazides, thiols, aminomethyl benzimidazole, phosphorous-based, nitrogen-based and heterocycles bidentate chelators, and can be monodentade, bidentade, and tridentade chelates [2,6,18,28,29]. 

The non-hydroxamate-based inhibitors open up a wide spectrum of affinities for the zinc ion from the catalytic site and new opportunities for targeting and inhibiting the active center [18,28]. They have weak Zn^2+^ chelating ability and the rates of severe side effects, such as the musculoskeletal syndrome (MSS) decreased dramatically compared with the hydroxamate inhibitors [28].

#### 4.2.1. Thiolates-Based Inhibitors

The ability of the monodentate binding of thiols to zinc ion in proenzymes has served as inspiration for the design of several MMPis [5,29]. The potency of thiol inhibitors is intermediate between that of hydroxamate- and carboxylate- based inhibitors [29]. The first example of inhibitor thiol-based for MMP-1 is a bipeptidic analogue, where the incorporation of a thiol group as α substituent leads to improvement of activity (Figure 19a) [19]. Derivates with “linker” substituent between P_1_-P_1_’ positions show a total loss of activity (Figure 19b) [19,62]. On the contrary incorporation of a methyl carboxylate group leads to a significant increase in activity (Figure 19c) [19]. The increased activity of these compounds may be a consequence of beneficial interactions between S_1_, the carbonyl ester, and the thiol group, participating in the bidentate coordination of the zinc [19]. 

Montana et al. have identified a series of inhibitors with mercaptoacyl, obtaining moderate inhibitors (Figure 20a) against a wide variety of enzymes with a deep pocket shown to be orally active in mouse models with arthritis [63]. The thiol and acyl carbonyl groups could cooperate in binding to the zinc of the active site [63]. Warshasky et al. have produced a variety of compounds in the Montana series, in which the amide nitrogen P_2_’ is linked to the group P_1_’ (Figure 20b) [19].

The β-mercaptoacilamide represented in Figure 21 is active against the MMP-9 in vitro and exhibits oral activity in rats [19]. The 4-alcoxy substituent of cyclohexane group improved the activity against all MMPs [19]. Replacement of the 4-ethoxy substituent with 4-propyloxy leads to a significant reduction in MMP-1 activity and improves selectivity for MMP-3 [19]. The equivalent cyclopentyl compounds are inactive [19]. The mercaptoamide is unstable in solution hence, to overcome this issue, Campbell and Levin have prepared a series of mercaptoalcohols and mercaptoketones inhibitors [64]. The mercaptoalcohols have exhibited modest activity against MMP-1, -3, and -9, while the equivalent mercaptoketones could be optimized to active broad-spectrum inhibitors [64].

In 2005, Hurst et al. [65] reported a series of mercaptosulphides inhibitors that targeted MMP-1 [65]. The structure–activity relationship indicates that the five-membered ring increases the stability of the inhibitor compared to the linear structure, which can be quickly oxidized and lose its potency [65].

Table 9 shows the IC_50_ and Ki values of some thiolates-based inhibitors [6,15,16,17,18,19,29,35,37,54,55].

#### 4.2.2. Carboxylates-Based Inhibitors

The carboxylic inhibitors are synthetic precursors of the more popular hydroxamates yet they are weaker zinc (II) ligands than hydroxamates [17,27] and monodentate chelate [27]. Carboxylic acid is present in several MMPis that contain large lipophilic groups, such as biphenyls, since they fit in the S_1_’ pocket [5,6]. These ZBGs are particularly appreciated for their high stability in vivo and their great positive effects on solubility, bioavailability, and selective properties [5,17]. The hydroxamate-based inhibitors are more potent in physiological conditions than carboxylate inhibitors, due to differences in acidity constants [29]. The carboxylate inhibitors bind more tightly to MMPs at low pH, while hydroxamate-based ones have a wider range of pH from 5 to 8 [29]. Fray et al. [66] compared the inhibition profiles of hydroxamates and carboxylic inhibitors (Figure 21a) and observed that the substitution of a carboxylate by a hydroxamate causes a 10-fold increase in potency of the inhibitor towards MMP-3 but decreases the selectivity against MMP-1, -2, -9, and -14 [66]. This effect is attributed to the fact that the strong zinc (II) affinity to the hydroxamic acid group is the main determinant of the binding energy, while in carboxylates this energy relies to a bigger extent on specific interactions with the specific pockets [66].

Hagmann et al. [67] described a series with *N*-carboxyalkyl group substituents, which presented inhibition for MMP-1, -2, and -3 [67] (Figure 21b). However, the substitution of the phenethyl group, in P_1_’ position, for a linear alkyl chain removes the inhibitory activity for MMP-1 but it does not affect the activity for MMP-2 and -3 [67]. A similar effect was achieved by the 4-substitution of the phenyl ring of the phenethyl group with a small linear alkyl group [67] (Figure 22b). A similar range of P3′ esters has been identified with “phthalamidobutyl” (Figure 22b), increasing activity against MMP-3 and further increasing selectivity [67].

The interaction of the P_1_′ biphenyl substituent with pocket S_1_′ is an important factor contributing to the binding of the inhibitor [19]. The X-ray structure of the acyclic compound with MMP-3 revealed an important interaction between the phenyl terminal of the biphenyl group and the side chain of histidine (His_224_) [19]. The carboxylic acids derived from “D-valine” have a selective inhibition for MMP-2 and -3 [19]. The 4-substitution of the biphenyl ring helped to increase potency compared to the unsubstituted analogue and also helped to improve the pharmacokinetic properties [19].

With the aim of development inhibitors with high selectivity for a single MMP, Wyeth published, in 2005, a series of biphenyl compounds with carboxylates sulphonamides (Figure 23a). These compounds were tested for the treatment of osteoarthritis and indeed presented selectivity against MMP-13 [18]. Wyeth research developed a series of carboxylic acids-based inhibitors, which were potent and selective against MMP-13, with the carboxylate function connected to a benzofuran via a biphenyl sulphonamide spacer (Figure 23b) [16]. The presence of a bulky substituent in the benzofuran 4-position resulted in a compound 100-fold more selective for MMP-13 over MMP-2 [16].

Tanomastat (Bayer) inhibits MMP-2, -3, -9, and -13 but not MMP-1 [6,29]. This inhibitor participated in clinical trials, proving tolerable and no serious MSS, but the efficacy was negative in small-cell lung cancer because the median overall survival of patients treated did not increase [6,29].

Table 10 shows the IC_50_ and Ki values of some carboxylates-based inhibitors [6,15,16,17,18,19,29,35,37,54,55].

#### 4.2.3. Phosphorus-Based Inhibitors

The capacity of the phosphoric group to reproduce the gem-diol intermediate during peptide hydrolysis was explored with different structures to obtain potent MMPis [5]. The phosphorus-based of the peptide-analogous inhibitors can be phosphonates/phosphonic acids, phosphoramidates, phosphonamidates, and phosphinates/phosphinic peptides [27]. The phosphinic acid (PO(OH)-CH_2_) mimics the transition state obtained in substrate degradation, where each oxygen atom can coordinate both the catalytic zinc and the catalytic Glu [6,19,27]. The phosphinic acids are monodentate chelates [27]. In contrast to hydroxamate compounds, the phosphinic compounds interact with both the primed and unprimed side of the catalytic site [17,27,35] due to the placement of the ZBG in the middle of the scaffold and not at its *N*- or *C*-terminal, as in the cases of hydroxamate and carboxylate inhibitors [17]. Another advantage of phosphinic acids is the improved metabolic stability compared with hydroxamate acids [27]. 

The effectiveness of phosphoric acid inhibitors has been studied and it has been found that the three pockets unprimed are connected to obtain the maximum performance [19]. The S_1_-S_2_ pockets can be exploited using aromatic groups [19,68], that is why Reiter et al. prepared compounds with 4-benzyl (Figure 24) as a substituent to fill S_2_ pocket. They found that in the absence of this substituent or its replacement by small aliphatic or cyclohexyl methyl groups led to a loss of activity [19,68].

Matziari et al. [69] synthesized a series of phosphinic pseudopeptides bearing long P_1_’ side chains, compounds that contain groups at the *ortho*-position of the phenyl ring and are selective for MMP-11 by the interaction of these groups with residues located at the entrance of the S_1_’ cavity [69]. These results suggest that the development of compounds able to probe the entrance of the S_1_’ cavity might represent an alternative strategy to gain selectivity [69].

Other phosphorus-based ZBGs are the carbamoyl phosphates, in which the two oxygens form a five membered ring with the zinc ion [18]. The negative charge of these inhibitors prevents their penetration into the cell and restrain them for extracellular space, contributing to low cytotoxicity [18]. Pochetti et al. [70] described a compound with high affinity to MMP-8 (Ki = 0.6 nM) but inhibits also MMP-2 (Ki = 5 nM) and MMP-3 (Ki = 40 nM) (Figure 25). The R enantiomer is more potent (1000 time more) than the S enantiomer (Ki = 0.7 μM) [70].

The classical approach to synthesizing phosphinic compounds limits the full exploitation of this class of compounds for development of highly selective inhibitors of MMPS [35]. 

Table 11 shows the IC_50_ and Ki values of some phosphorus-based inhibitors [6,15,16,17,18,19,29,35,37,54,55].

#### 4.2.4. Nitrogen-Based Inhibitors

The nitrogen-based inhibitors have a binding preference to late transition metals and improved selectivity to zinc-dependent enzymes like MMPs [2]. The nitrogen-based inhibitors are studied by the Food and Drug Administration (FDA) and its metabolic availability and bioavailability are well described [2,18]. This ZBG type binds to Zn^2+^ using the nitrogen atom and the carbonyl oxygen adjacent to nitrogen, which favors the formation of an enol because it is established by two hydrogen bonds [18]. 

The pyrimidine-2,4,6-trione inhibitors were published in 2001 by Hoffman-LaRoche. These compounds show relative specificity to gelatinases and potential usefulness as anticancer drugs [2]. The derivatization of position 5 of this compound promotes access to S_1_’ and S_2_’ pockets [18]. In development of osteoarthritis drugs, the pyrimidine-2,4,6-trione inhibitors have been optimized to inhibit MMP-13 [2]. 

#### 4.2.5. Heterocyclic Bidentate-Based Inhibitors

Heterocyclic bidentate ZBGs have better biostability and higher catalytic zinc ion binding capacity than hydroxamic acids, due to ligand rigidity [2]. Compared heterocycles bidentate and acetohydroxamic acid, the first are more potent to inhibit MMP-1, -2, and -3 and show low toxicity in cell viability assays [2].

Pyrones are biocompatible and they present good aqueous solubility [16]. The arylic portion was added to fit the MMP-3 hydrophobic S_1_’ pocket, resulting in the compounds more potent than the corresponding hydroxamate-based inhibitors [16].

Table 12 shows the IC_50_ and Ki values of some heterocyclic bidentate-based inhibitors [6,15,16,17,18,19,29,35,37,54,55].

#### 4.2.6. Tetracyclines-Based Inhibitors

Tetracyclines are antibiotics that can chelate zinc and calcium ions and inhibit MMP activity [2,16,29]. Chemically modified tetracyclines (CMT) are preferred over conventional tetracyclines because they reach higher plasma levels for prolonged periods, consequently require less frequent administration, cause less gastrointestinal side effects, and have promising anti-proliferative and anti-metastatic activity [2,16,29]. The CMT binds to pro- or active MMPs, disrupt the native conformation of the protein, and leave the enzymes inactive [29]. In the search for new anticancer agents, the first series of CMT was obtained by removal of the dimethylamino group from the carbon-4 position, resulting in a compound without antimicrobial activity but with anticollagenolytic activity, in vitro and in vivo [16]. Preclinical studies demonstrated that CMT can inhibit gelatinases, stromelysins, collagenases, and MT-MMPs, by downregulating the expression of gelatinases, reducing the production of pro-enzymes and inhibiting the activation of pro-gelatinases and pro-collagenases [16,29].

Doxycycline (Figure 26a) is a semi-synthetic tetracycline that inhibits MMP-2, -9, -7, and -8 and is the only compound approved as an MMP inhibitor for the treatment of periodontitis [2,6]. The COL-3 (Figure 26b) showed specificity for MMP-2, -9, and -14, by decrease trypsinogen-2 and inducible nitric oxide (iNO) production, which are regulators of MMP activity [16]. Although COL-3 is currently being evaluated in clinical phase II trials, it showed poor solubility and stability [16].

Table 13 shows the IC_50_ values of some tetracyclines-based inhibitors [6,15,16,17,18,19,29,35,37,54,55,71].

#### 4.2.7. Mechanism-Based Inhibitors

The mechanism-based inhibitors coordinate with catalytic zinc ion, in a monodentate mode, allowing the nucleophilic attack by a conserved glutamic residue on the active site and forming a covalent bond [2,17]. This attack causes a conformational change in the catalytic site environment [17] preventing dissociation of the inhibitor and decreasing the rate of catalytic turnover and the amount of inhibitor needed to saturate the enzyme [2].

In 2000, Mobashery et al. [72] were the first to report this novel type of MMPi that blocks gelatinases with a unique mechanistic mode [72]. The thiirane inhibitor showed a mechanism-based, slow-binding inhibition for MMP-2 and MMP-9 [72]. Bernardo et al. [73] also reported a slow-binding thiirane-containing inhibitor, (Figure 27), selective for MMP-2 and -9, where the sulfur group coordinates with the catalytic zinc ion, activates the thiirane group to interact with the active site glutamate, by nucleophilic attack causing a loss of activity [73]. These inhibitors are the first example of a suicide-inhibitor of MMPs [73].

Thiirane-based ND-322 is a small molecule selective to MMP-2/MT1-MMP [2]. This inhibitor has been shown to reduce melanoma cell growth, migration, and invasion, and to delay metastatic dissemination [2]. 

SB-3CT is a selective inhibitor of MMP-2 and -9 [2]. The inhibition mechanism is similar to a “suicide inhibitor” in which a functional group is activated, leading to covalent modification of the active site [2]. SB-3CT also shows slow-binding kinetics with MMP-2, -3, and -9, contributing to slow dissociation of the MMP-inhibitor complex, but it is a reversible inhibitor which differentiates it from the truly irreversible suicide inhibitors [2]. O SB-3CT has potential benefits in brain damage caused by cerebral ischemia and has anti-cancer effects in T-cells lymphoma and prostate cancer models [2]. 

### 4.3. Catalytic Domain (Non-Zinc Binding) Inhibitors

The catalytic domain of MMPs contains other regions that can be exploited [17]. The first 3D-structure of the complex MMP-1 (catalytic domain)-synthetic inhibitor was reported in 1994 by Glaxo researchers [35]. Thereafter, other complexes have been studied and it was found that the S_1_’ pockets have different depths among MMPS and this difference has been utilized in developing selective MMPis [28,35]. 

Stockman and Finel optimized two distinct series of MMP-3 inhibitors: PNU-141803 (amide, Figure 28a) and PNU-142372 (urea, Figure 28b) [19]. The connection between MMP-3 and PNU-142372 shows that the aromatic ring from the inhibitor extends to the S_3_ pocket (hydrophobic) and the thiadiazole sulfur group interacts with the catalytic zinc [19]. Moreover, the two nitrogen atoms form hydrogen bonds with Ala_164_ and Glu_202_ residues [19]. The alkylation of nitrogen atom or its replacement for carbon leads to the removal activity [19]. The replacement of a tyrosine for a serine within the S_3_ pocket (present in MMP-1) leads to the removal of inhibitory activity and explains the absence of activity against collagenases [19].

Sanofi-Aventis developed a compound (Figure 29) for MMP-13 (IC_50_ = 6.6 μM), with very high selectivity [6]. This compound binds deeply to the S_1_’ pocket and to a side pocket that has not been identified for other MMPs [6]. The pyridyl moiety is towards to the entrance of the S_1_’ pocket, without interacting with the catalytic Zn(II) ion and the oxygen atoms neither from the amide (peptidic) bonds of the main chain (between Thr_245_ and Thr_247_) nor from hydroxyl group from the Thr_247_ side chain in the S_1_’ pocket [6].

Many natural compounds have been shown to possess selective inhibition [28]. Wang et al. identified 19 potential MMPis from 4000 natural compounds isolated from medicinal plants [28]. The caffeates and flavonoids were found to be selective inhibitors against MMP-2 and -9, by occupying the S_1_’ and S_3_ pockets [28].

The marine natural products are another pharmacological resource and include derivates from algae, sponges, and cartilages [28]. Some examples are Neovastat, Dieckol, and Ageladine A and they manifest anti-angiogenic, anti-proliferative, and anti-tumor effects [28].

Although the natural MMPis are more biocompatible and less toxic, they have disadvantages such as the effective dosages are in micromolar scale, which is thousands of times higher than synthetic inhibitors and are difficult to patent, making the pharmacological companies and investors reluctant to sponsor large-scale clinical trials [28].

### 4.4. Allosteric and Exosite Inhibitors

The catalytic zinc ion is common in all MMPs, therefore, if interactions of the substrates with this ion are minimized this would improve the inhibitor selectivity [2]. The hemopexin-like domain can move relatively to the catalytic domain and allosterically manipulate enzymatic activity by conformation deformation [28]. The allosteric drugs have a non-competitive inhibition mode [16,28], they bind and lock the MMP active site, forcing it to take less favorable conformation for substrate binding [2,16], avoiding off-target inhibition [28] and preventing the occurrence of side effects [28]. Exosite inhibitors are another alternative for selective MMPis since these inhibitors bind to alternative sites of MMPs [16,28].

Remacle et al. reported NSC405020, a small molecule that binds selectively to the hemopexin-like domain of MMP-14 [28]. This molecule inhibits the MMP-14 homodimerization and the interaction between the hemopexin-like domain and catalytic domain, preventing the type I collagen degradation [28].

Dufour et al. developed a peptide targeting the MMP-9 hemopexin-like domain, which blocks MMP-9 dimer formation and cell migration [28]. Scannevin et al. identified a highly selective compound (JNJ0966), which binds to the MMP-9 pro-peptide domain, inhibiting its activation, but not affecting the activity of MMP-1, -2, and -14 [28].

Xu et al. synthesized a peptide which inhibits the hydrolysis of type I and IV collagen by MMP-2, through binding to its collagen binding domain [28].

### 4.5. Antibody-Based Inhibitors

Antibodies are selective and have high affinity to MMP [27]. Clinical trials utilizing antibodies have provided evidence that selective MMP inhibitors do not induce MSS [27]. The therapeutic potential of anti-MMP antibodies has yet to be realized [27]. The antibodies may also undergo proteolysis, may be removed from circulation rapidly, and are costly [27]. 

Antibodies are large Y-sharped proteins which bind to an antigen via the fragment antigen-binding (Fab variable region) [28]. Monoclonal antibodies are highly specific, and they have affinity to MMPs [2,29]. The hemopexin domain can be a potential target for MMPs antibodies [2].

REGA-3G12 and REGA-2D9 are antibodies specific to MMP-9 [2,27,28] but not MMP-2 [28]. The inhibition mode of the REGA-3G12 involves the catalytic domain, the *N*-terminal region Trp_116_-Lys_214_ [28], and not the catalytic zinc ion or the fibronectin region [2]. The AB0041 and AB0046 are monoclonal anti-MMP-9 antibodies, which showed inhibition to tumor growth and metastasis in a model of colorectal carcinoma [27]. 

Andecaliximab (GS-5745), the humanized version of AD0041 [27], is a highly selective antibody and exerts allosteric control over tumor growth and metastasis in a colorectal carcinoma model (IC_50_ = 0.148 nM) [28]. This antibody is the only inhibitor that has undergone clinical trials [27,28] and it inhibits the pro-MMP-9 activation and inhibits non-competitively the MMP-9 activity [27].

DX-2400 is an antibody isolated from phage and it targets MMP-14 and the MMP-14-pro-MMP-2 complex, decreasing MMP activity [28]. DX-2400 inhibits the metastasis in a breast cancer xenograft mouse model [27]. However, the LAM-2/15 is the only selective inhibitor that inhibits MMP-14 catalytic activity, but not the pro-MMP-2 activation or MMP-14 dimerization [28]. The 9E8 is another antibody targeting MMP-14 which does not affect the catalytic activity of MMP-14 but inhibits the pro-MMP-2 activation [28].

Human scFv-Fc antibody E3 is bound to the catalytic domain of MMP-14 and inhibits type I collagen binding [27]. Human antibody Fab libraries were synthetized and the peptide G sequence (Phe-Ser-Ile-Ala-His-Glu) was incorporated resulting in Fab 1F8 antibody inhibitor, which inhibits the catalytic domain of MMP-14 [27].

Antibodies can also inhibit a specific activation of an MMP [27]. For example, the mAb 9E8 inhibits the MMP-14 activation of proMMP-2 but not the catalytic activities of MMP-14 [27]. The antibody LOOP_AB_ also inhibits the MMP-14 activation of pro-MMP-2 but not collagenolysis activity of MMP-14 [27]. There are antibodies that reduce the MMP expression [71].

## 5. Why Do MMP is Fail?

MMI inhibitor side effects are predominantly related to off-target metal chelation [74]. The majority of MMPis used clinically are hydroxamic acid derivates with low selectivity [74] hence, they can inhibit other proteinases [28]. The most frequent side effects observed in MMPis clinical trials is the musculoskeletal syndrome (MSS) [17,28], which manifested as pain and immobility in the joints, arthralgias, contractures in the hands, and an overall reduced quality of life [14,15,74], leading to MMPis failing in the last phases of clinical trials [14]. Several studies indicate that the development of MSS is related to dose and time, with slightly different kinetic for the different MMPis, and the development of MSS is an indicator of successful MMPis [14,74]. The MMP inhibitors focused on chelating the catalytic zinc ion have poor selectivity and resulted in MSS and gastrointestinal disorders [27]. However, the exact causes of MSS remains unknown [74], but can be related to a simultaneous inhibition of several MMPs [6,17,27].

Analysis of the expression of a target protein shows its presence at high levels when a disease is manifested or at low levels or absence in a healthy state [74]. However, these studies do not determine if a particular protein is directly associated with the disease process or if it is involved in ancillary event [74]. Studies of genetic manipulation in mouse as animal models determine the roles of MMPs in various pathological processes [74]. However, there are caveats in the use of animal models [74]:-The observed effects can be a consequence of the manipulated absence of MMP, being a compensation mechanism;-The mouse models are unable to replicate the complexity of any human disease. The mouse models serve to recreate specific processes or sets of processes but not the physiological changes that occur in humans.

## 6. Conclusions

Due to the side effects rising from the lack of selectivity and from the insufficient knowledge about the role of each MMP in the different pathological processes, none of the designed synthetic MMP inhibitors have yet passed the clinical trials and reached the market [6,27]. The poor performance of MMP inhibitors in clinical trials has globally been attributed to [27]:-Inhibition of other metalloenzymes;-Lack of specificity within the MMP family;-Poor pharmacokinetics;-Dose-limiting side effects/toxicity;-In vivo instability;-Low oral availability/inability to assess inhibition efficacy.

In 1988, the first inhibitor was synthesized but after nearly 30 years, only one drug, Periostat^®^, doxycycline hydrate, had obtained approval from the FDA for the treatment of periodontal disease [6,16,17,27,28]. This inhibitor exhibited also therapeutic effects in treating aortic aneurysm, multiple sclerosis, as well as Type II diabetes [28].

## Figures and Tables

**Figure 1 biomolecules-10-00717-f001:**
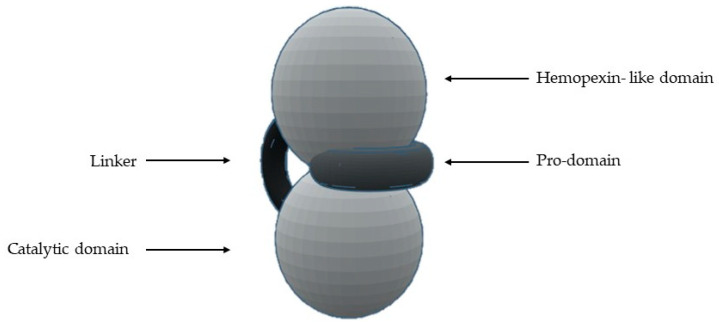
Schematic representation of the general structure of MMP.

**Figure 2 biomolecules-10-00717-f002:**
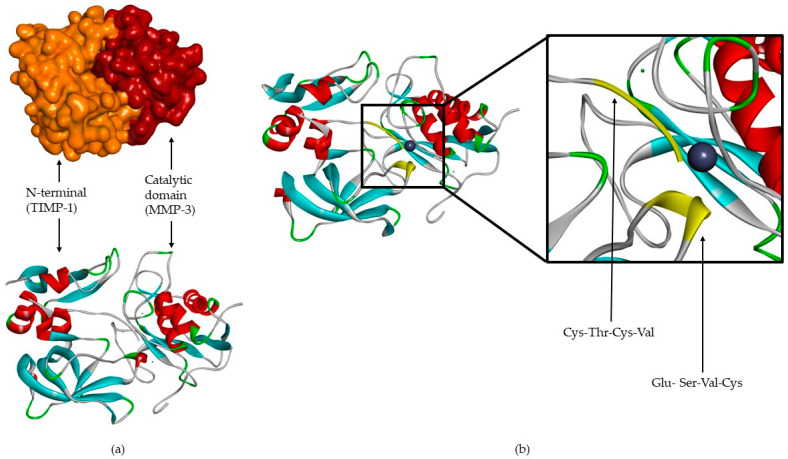
(**a**) TIMP-1-catalytic domain of the MMP-3 complex. (**b**) TIMP-1-catalytic domain of the MMP-3 complex, where two conserved groups, Cys-Thr-Cys-Val and Glu-Ser-Val-Cys, are represented in yellow.

**Figure 3 biomolecules-10-00717-f003:**
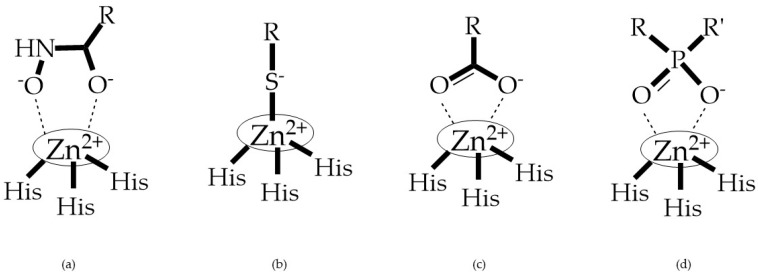
Examples of zinc binding groups (ZBGs). (**a**) Hydroxamate-based inhibitor; (**b**) thiolate-based inhibitor; (**c**) carboxylate-based inhibitor; (**d**) phosphorous-based inhibitor. The R group is the scaffold of inhibitor.

**Figure 4 biomolecules-10-00717-f004:**
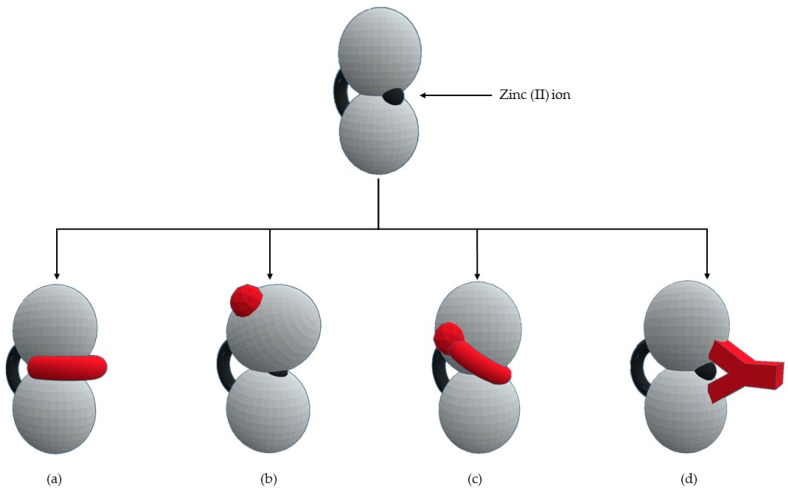
Types of MMPs’ inhibitors: (**a**) endogenous-like inhibitors that chelate the catalytic zinc (II) ion; (**b**) exosite targeting inhibitors that alter the conformation of the enzyme; (**c**) a combination of exosite binding and metal chelating inhibitors; (**d**) antibodies inhibitors.

**Figure 5 biomolecules-10-00717-f005:**
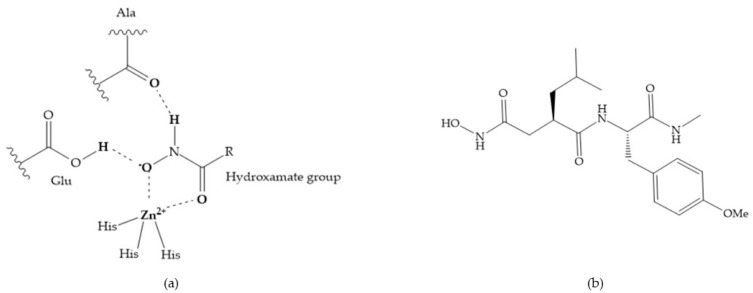
(**a**) Interaction between hydroxamate group and catalytic zinc (II) ion. The oxygen of the hydroxamate forms a strong hydrogen bond with the carboxylate oxygen of the catalytic Glu, while the NH of hydroxamate establishes another hydrogen bond with the carbonyl oxygen of Ala; (**b**) S*C*-44463 inhibitor.

**Figure 6 biomolecules-10-00717-f006:**
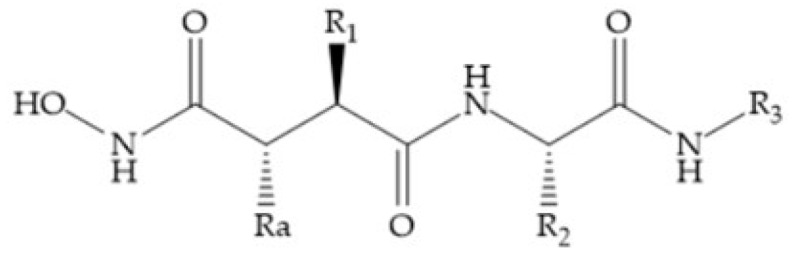
General structure of hydroxamic acid. OH group: catalytic zinc binding group; R_a_: α substituent; R1: P_1_’ substituent group and this group is determinant to selectivity and activity; R_2_: P_2_’ substituent and this substituent can be cyclized with R_a_ and R_3_; R_3_: P_3_’ substituent.

**Figure 7 biomolecules-10-00717-f007:**
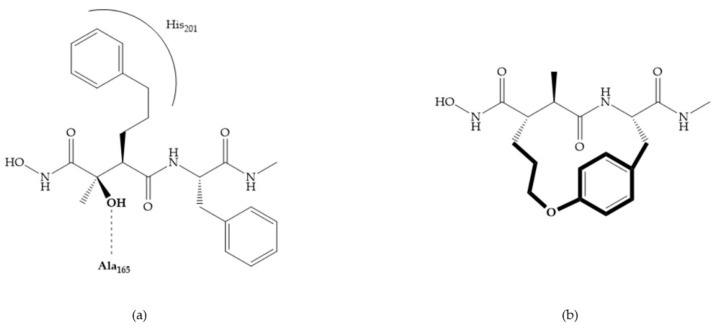
(**a**) Analogue of Marimastat with the α position disubstituted; (**b**) analogue of Marimastat with Ra and R_2_ position connected; (**c**) SE205; (**d**) SC903.

**Figure 8 biomolecules-10-00717-f008:**
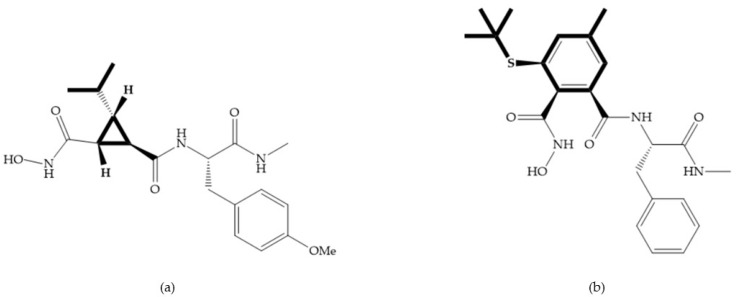
Inhibitors with conformational restrictions between Ra and R_1_ positions. (**a**) Inhibitor with three-membered ring; (**b**) inhibitor with six-membered ring.

**Figure 9 biomolecules-10-00717-f009:**
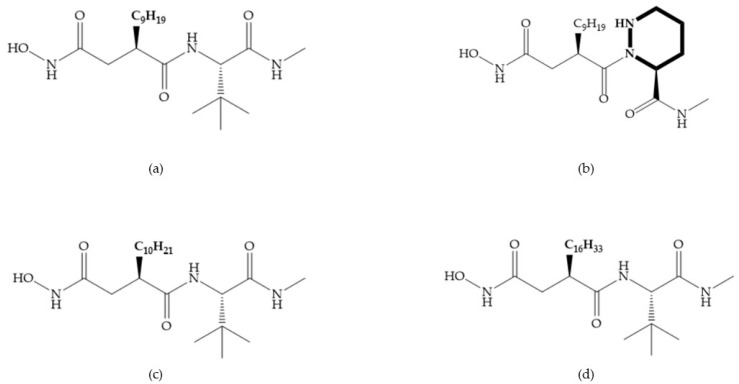
Inhibitors with modification of R_1_ position. (**a**) Inhibitor with alkyl chain. This inhibitor has activity against MMP-2, -3, and -9, but the inhibition of MMP-1 is low; (**b**) R-94138, Matlystatin derivate. The inhibition of MMP-9 is 10 times higher than analogues with C_8_ or C_10_ chains; (**c**) succinyl hydroxamate analogue with C_10_ chain, which inhibits MMP-1; (**d**) succinyl hydroxamate analogue with C_16_ chain, which inhibits MMP-1.

**Figure 10 biomolecules-10-00717-f010:**
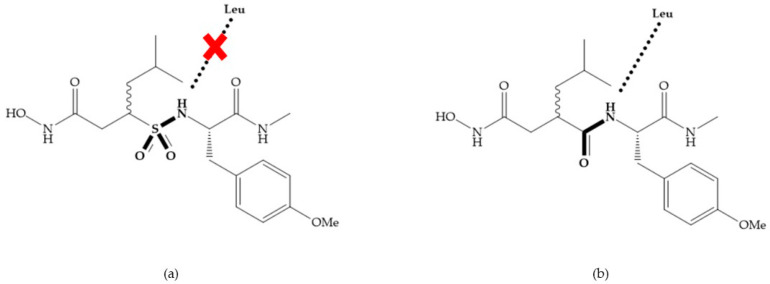
(**a**) Succinyl hydroxamate acid with a sulphonamide bond. This compound presents low inhibitory activity because of the pyramidal nature of the sulphonamide group. (**b**) Succinyl hydroxamate acid with carbonyl bond.

**Figure 11 biomolecules-10-00717-f011:**
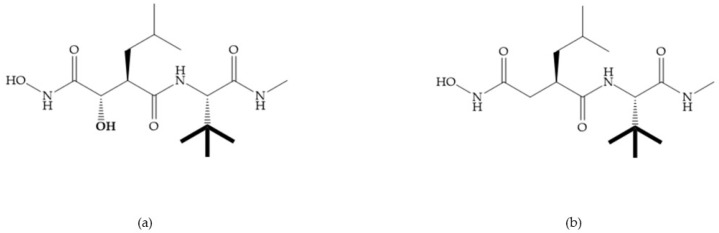
(**a**) Marimastat. The Ra position is substituted with a hydroxyl group (OH). (**b**) Ro31-9790. The Ra position has no substituents.

**Figure 12 biomolecules-10-00717-f012:**
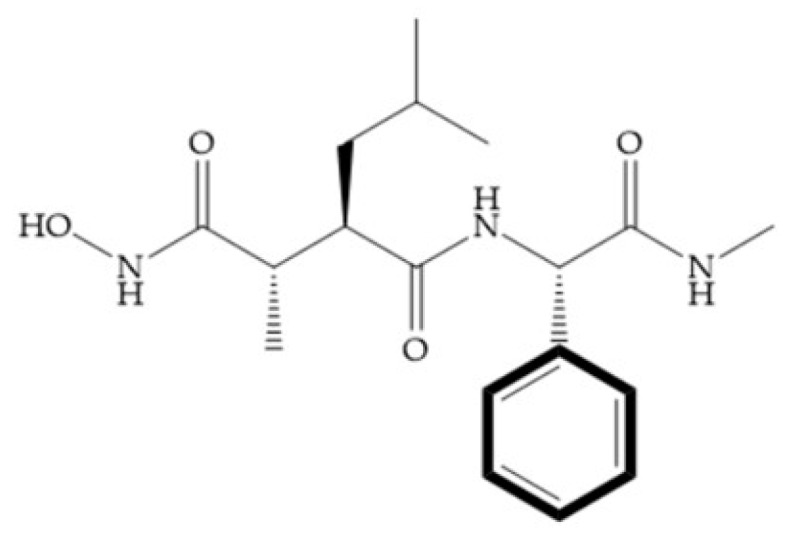
KB-R7785 inhibitor.

**Figure 13 biomolecules-10-00717-f013:**
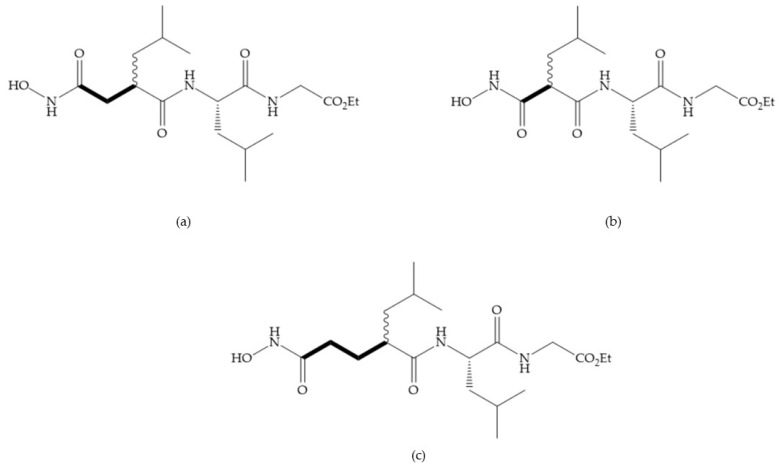
(**a**) Derivates of succinyl hydroxamic acid; (**b**) malonyl acid; (**c**) glutaryl acid.

**Figure 14 biomolecules-10-00717-f014:**
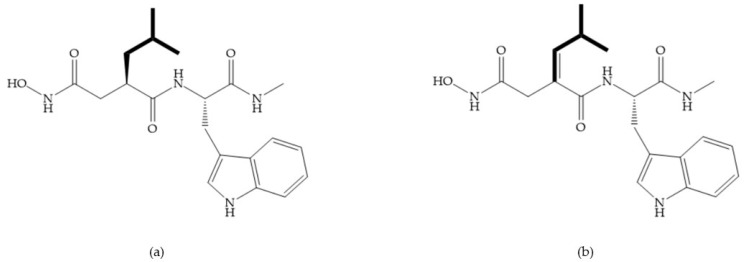
(**a**) Ilomastat; (**b**) Ilomastat derivate with isobutylidene group.

**Figure 15 biomolecules-10-00717-f015:**
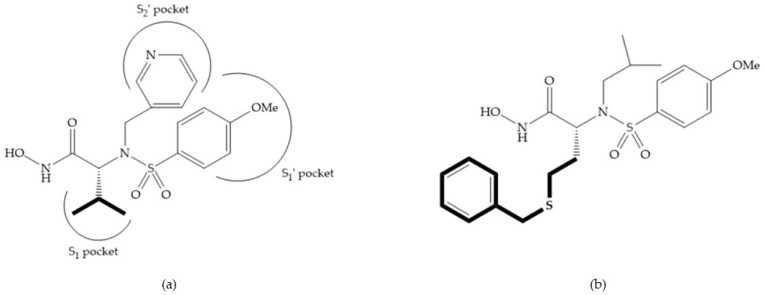
(**a**) CGS-27023A; (**b**) thioester derivate of CGS-27023A.

**Figure 16 biomolecules-10-00717-f016:**
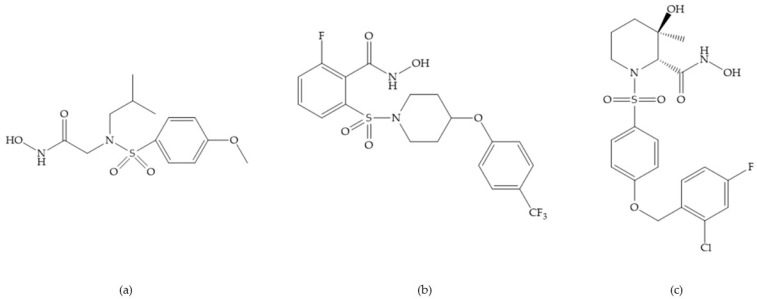
(**a**) NNGH; (**b**) arylhydroxamate sulphonamide compound; (**c**) 3-hydroxy-3-methylpipecolic hydroxamates.

**Figure 17 biomolecules-10-00717-f017:**
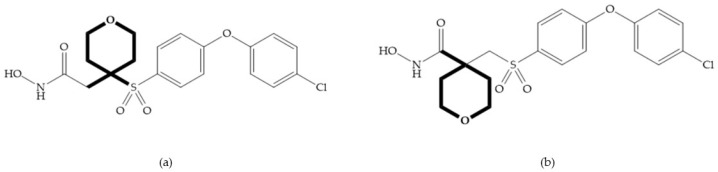
(**a**) RS-113,456; (**b**) RS-130,830.

**Figure 18 biomolecules-10-00717-f018:**
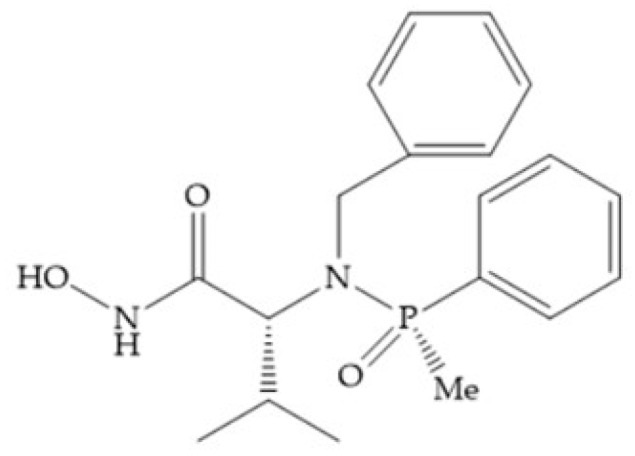
Inhibitor with phosphinamide group.

**Figure 19 biomolecules-10-00717-f019:**
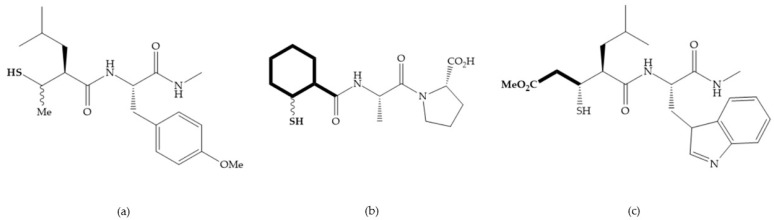
(**a**) Thiol-based inhibitor with the thiol group as α-substituent. The stoichiometric is S when the thiol group is present. In its absence, the compound with R stoichiometric is more active than the S analogue; (**b**) thiol-based inhibitor with “linker” substituent; (**c**) thiol-based inhibitor with methyl carboxylate group.

**Figure 20 biomolecules-10-00717-f020:**
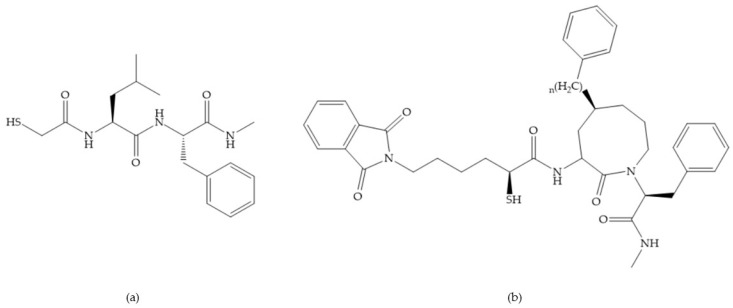
(**a**) Inhibitor with mercaptoacyl, where the thiol and acyl carbonyl groups could cooperate in binding to the zinc ion. (**b**) Variant of Montana compounds. n = 0, the compound does not have activity against MMPs-2, 3, and -12. n = 1, the compound has low activity against MMP-3.

**Figure 21 biomolecules-10-00717-f021:**
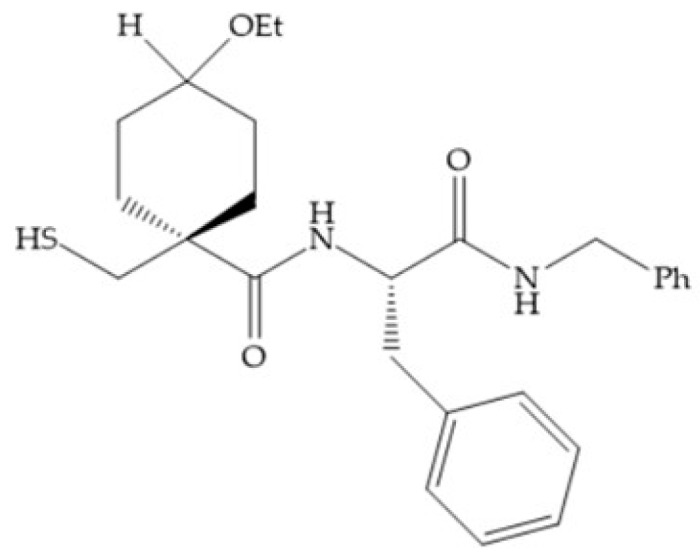
β-mercaptoacilamide inhibitor.

**Figure 22 biomolecules-10-00717-f022:**
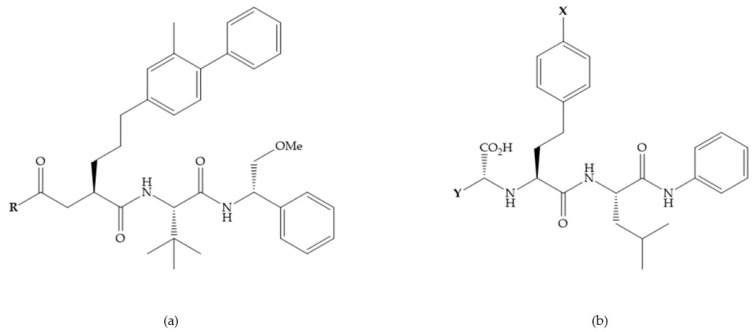
(**a**) Fray et al. inhibitors. R = NH(OH), hydroxamate-based inhibitor. R = OH, carboxylate-based inhibitor. (**b**) Hagmann inhibitors. If X = H and Y = Me the compound presents inhibition to MMP-1, -2, and -3. When X = C_4_H_9_ and Y = Me, the inhibitor has a similar effect to the previous one. The inhibitor with X = H and Y = Phthbutyl (phthalamidibutyl) shows activity against MMP-3.

**Figure 23 biomolecules-10-00717-f023:**
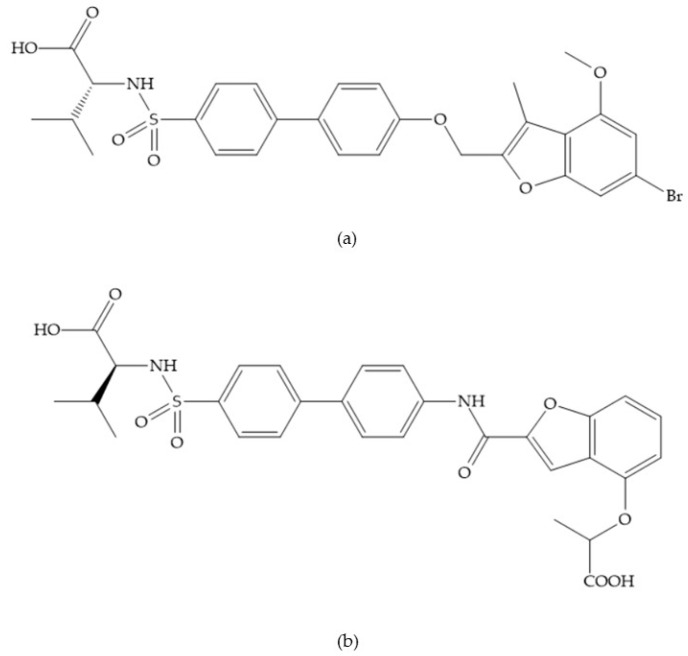
(**a**) Inhibitor of Wyeth with carboxylate sulphonamide; (**b**) inhibitor of Wyeth with the carboxylate function connected to a bezofuran via biphenyl sulphonamide spacer.

**Figure 24 biomolecules-10-00717-f024:**
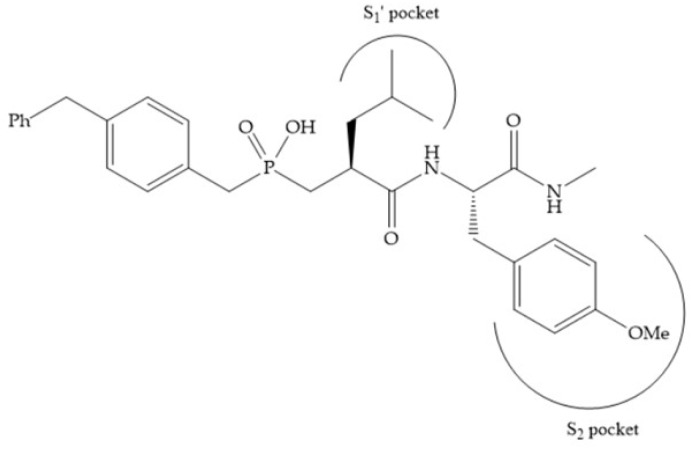
Compound prepared by Reiter et al., with a 4-benzyl substituent. The 4-benzyl group fills the S_2_ pocket and if the benzyl group was omitted or replaced by a small aliphatic or cyclohexyl methyl group, the activity is lost. The isobuthyl group fills the S_1_’ pocket in a manner similar to other substrate-like inhibitors.

**Figure 25 biomolecules-10-00717-f025:**
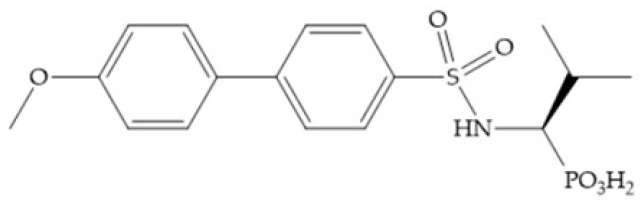
Pochetti’s inhibitor.

**Figure 26 biomolecules-10-00717-f026:**
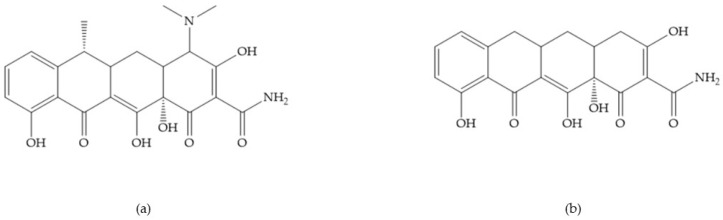
(**a**) Doxycycline; (**b**) COL-3.

**Figure 27 biomolecules-10-00717-f027:**
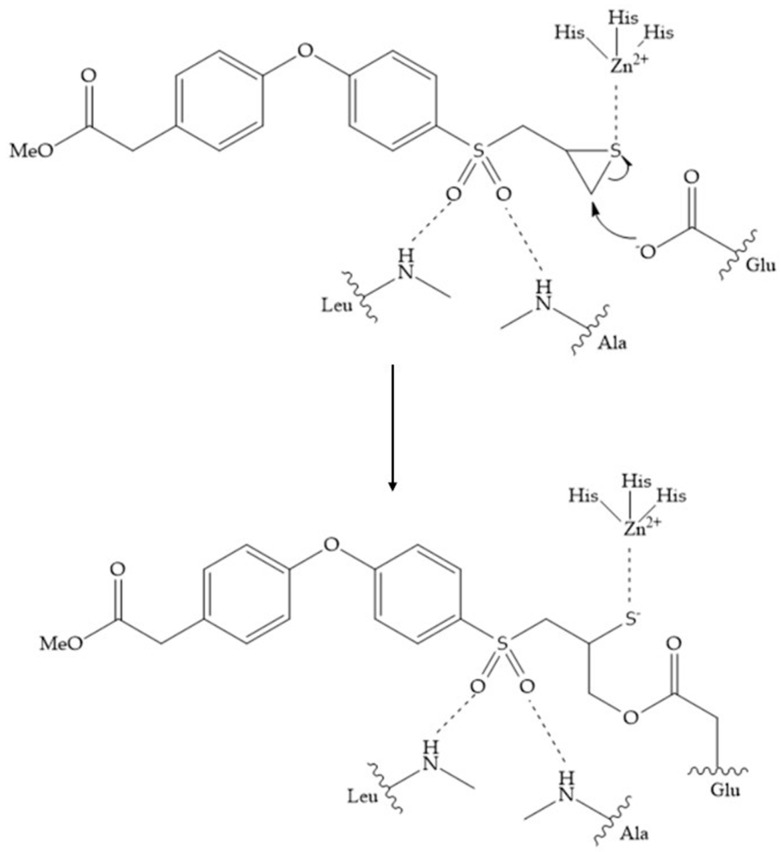
Inhibitor developed by Bernardo el al. The sulfur group coordinates with the catalytic zinc ion and the activation of the thiirane group happened with interactions between the active site glutamate, by nucleophilic attack.

**Figure 28 biomolecules-10-00717-f028:**
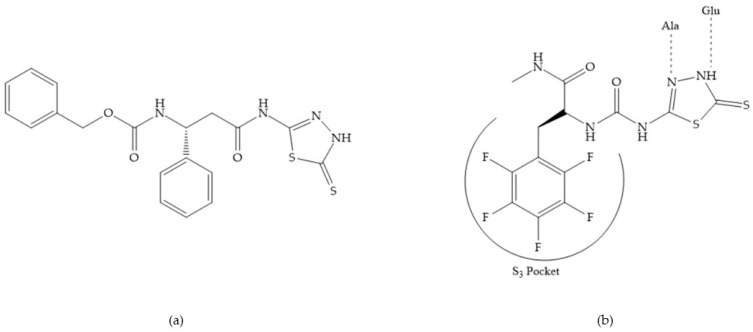
(**a**) PNU-141803; (**b**) PNU-142372.

**Figure 29 biomolecules-10-00717-f029:**
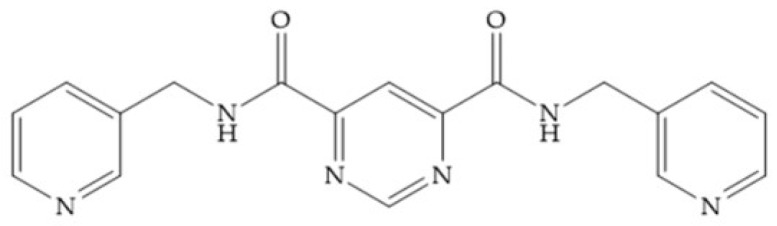
Sanofi-Aventis compound.

**Table 1 biomolecules-10-00717-t001:** Matrix metalloproteinases (MMPs) classes.

Class	MMP	
Collagenases	MMP-1, Collagenase-1, Interstitial or Fibroblast collagenases	
	MMP-8, Collagenase-2, or Neutrophil collagenases	
	MMP-13 or Collagenase 3	
Gelatinases	MMP-2 or Gelatinase A	
	MMP-9 or Gelatinase B	
Stromelysin	MMP-3 or Stromelysin-1	
	MMP-10 or Stromelysin-2	
	MMP-11	
Matrilysin	MMP-7	
	MMP-26, Matrilysin-2, or Endometase	
Membrane-type	Type I transmembrane protein	MMP-14 or MT1-MMP
		MMP-15 or MT2-MMP
		MMP-16 or MT3-MMP
		MMP-24 or MT5-MMP
	Glycosylphosphatidylinositol (GPI)-anchored	MMP17 or MT4-MMP
		MMP-25 or MT6-MMP
Other MMPs	MMP-12	
	MMP-19	
	MMP-20	
	MMP-21	
	MMP-23	
	MMP-27	
	MMP-28	

**Table 2 biomolecules-10-00717-t002:** MMPs inhibitors classification.

	Specific Inhibitor	Tissue Inhibitor of Metalloproteinases (TIMP)
Endogenous inhibitor	Non-specifics inhibitors	α2-macroglobulin
		Tissue factor pathway inhibitor (TFPI)
		Membrane-bound β-amyloid precursor protein
		*C*-terminal proteinases enhancer protein
		Reversion-inducing cystein-rich protein with Kasal domain motifs (RECK)
		GPI-anchored glycoprotein
Synthetic inhibitor	Hydroxamate-based inhibitors	
	Non-hydroxamate-based inhibitors	
	Catalytic domain (non-zinc binding) inhibitors	
	Allosteric and exosite inhibitors	
	Antibody-based inhibitors	

**Table 3 biomolecules-10-00717-t003:** Tissue inhibitors of metalloproteinases (TIMPs) classification.

TIMP	Expression	Inhibition	Inhibition Mode
**1**	Several tissues with transcription inducible by cytokines and hormones	Strong interaction with MMP-1, -2, -3, and -9 Weak interaction with MT1-MMP, MT3-MMP, MT5-MMP, and MMP-19	TIMP-1 forms a complex with pro-MMP-9 by binding to the hemopexin domain
**2**	Constitutive expression	Strong interaction with MMP-2	TIMP-2 has four residues in the *N*-terminal domain and an adjacent CD-loop region, which allows interaction between TIMP and the active center of MMP-2
**3**	In response to mitogenic stimulation and during cell cycle progression	MMP-1, -2, -3, -9, and -13	The inhibition mode is different from the other TIMPs for its unusual localization, as it is largely sequestered into the extracellular matrix or at the cell surface via heparan sulphate proteoglycans
**4**	Especially abundant in the heart, but is also expressed in injured tissue	MMP-2 and -14	-

**Table 4 biomolecules-10-00717-t004:** Non-specific endogenous inhibitors [4,7,12,13,33,34].

Non-Specific Inhibitor	Inhibition
α2-macroglobulin	MMP-2 and -9
Tissue factor pathway inhibitor	MMP-1 and -2
Membrane-bound β-amyloid precursor protein	MMP-2
*C*-terminal proteinase enhancer protein	MMP-2
Reversion-inducing-cysteine-rich protein with Kasal motifs (RECK)	MMP-2, -9, and -14
GPI-anchored glycoprotein	-

**Table 5 biomolecules-10-00717-t005:** Batimastat and Marimastat.

Name	Molecule	α Substituent	Effect
Batimastat	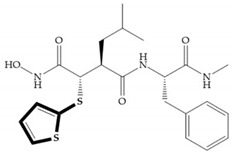	Thienylthiomethylene	Not available orally
Marimastat	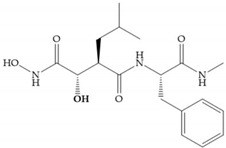	Hydroxyl group (directed to the protein surface, allowing the formation of hydrogen bonds with solvent)	Available orally

**Table 6 biomolecules-10-00717-t006:** IC_50_ and Ki values of succinyl hydroxamic acid-based inhibitors.

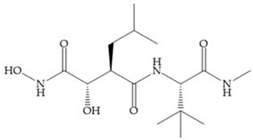 Marimastat (BB-2516) **IC_50_**: MMP-1 = 5 nM; MMP-2 = 6 nM; MMP-3 = 200 nM; MMP-7 = 20 nM; MMP-8 = 2 nM; MMP-9 = 3 nM; MMP-12 < 5 nM; MMP-13 = 0.74 nM; MMP-14 = 1.8 nM	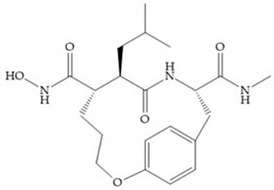 Analogue of Marimastat **IC_50_**: MMP-1 = 3 nM; MMP-2 = 1.1 nM; MMP-3 = 14 nM; MMP-7 = 0.8 nM; MMP-8 = 0.5 nM; MMP-9 = 0.9 nM; MMP-10 = 0.45 nM; MMP-14 = 94 nM; MMP-15 = 20 nM; MMP-16 = 35 nM			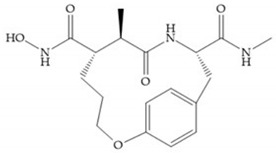 SE205 **Ki**: MMP-1 = 1.2 nM; MMP-3 = 32.7 nM; MMP-9 = 1.8 nM
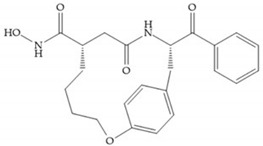 **IC_50_**: MMP-1 = 131 nM; MMP-2 = 9.5 nM; MMP-3 = 8.9 nM; MMP-7 = 3.3 nM	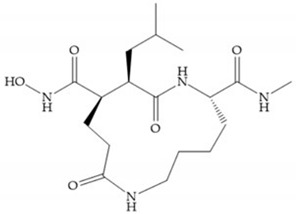 SC903 **Ki**: MMP-1 = 2.8 nM; MMP-3 = 24.1 nM; MMP-9 = 2.6 nM			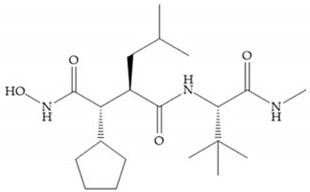 **IC_50_**: MMP-1 = 4 nM; MMP-2 = 3 nM; MMP-3 = 30 nM; MMP-7 = 20 nM; MMP-8 = 20 nM; MMP-9 = 9 nM
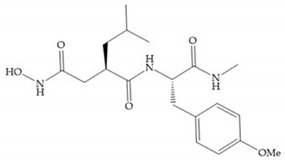 S*C*-44463 **IC_50_**: MMP-1 = 20 nM; MMP-2 = 6 nM; MMP-3 = 30 nM; MMP-7 = 30 nM	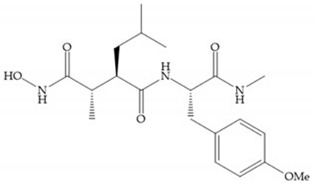 BB-16 **IC_50_**: MMP-1 = 5 nM; MMP-2 = 10 nM; MMP-3 = 40 nM; MMP-7 = 60 nM; MMP-8 = 7 nM			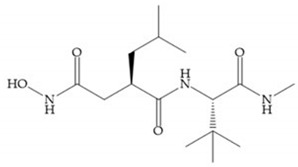 Ro-31-9790 **IC_50_**: MMP-1 = 10 nM; MMP-2 = 8 nM; MMP-3 = 700 nM; MMP-14 = 1.9 nM
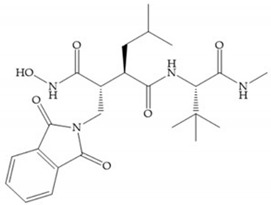 Ro-32-0554 **IC_50_**: MMP-1 = 0.5 nM; MMP-3 = 9.1 nM; MMP-9 = 4.3 nM	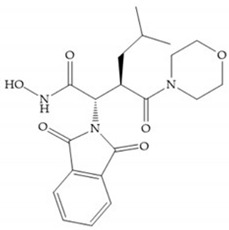 **IC_50_**: MMP-1 = 10 nM; MMP-2 = 400 nM; MMP-3 = 4.5 μM			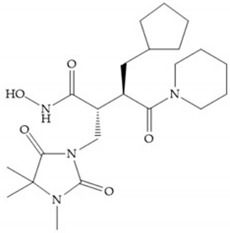 Ro-32-3555 **Ki**: MMP-1 = 3 nM; MMP-2 = 154 nM; MMP-3 = 527 nM; MMP-8 = 4 nM; MMP-9 = 59 nM; MMP-13 = 3 nM
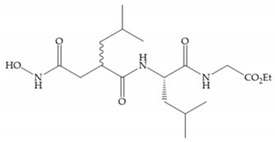 **IC_50_**: MMP-1 = 40 nM	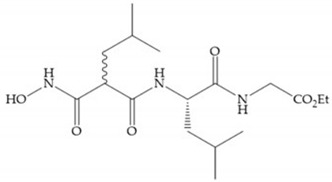 **IC_50_**: MMP-1 = 29 μM			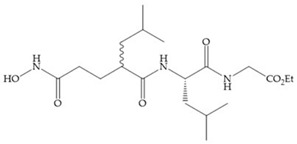 **IC_50_**: MMP-1 = 10 μM
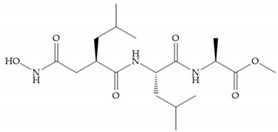 **IC_50_**: MMP-1 = 9 nM	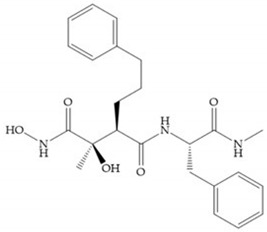 Analogue of Marimastat **IC_50_**: MMP-1 = 1 μM; MMP-2 = 15 nM; MMP-3 = 500 nM; MMP-7 = 10 μM; MMP-8 = 30 nM; MMP-9 = 15 nM			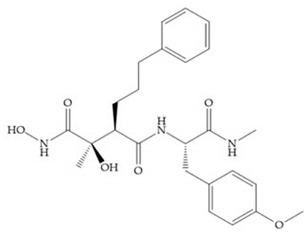 **Ki**: MMP-1 = 2 nM; MMP-3 = 3 nM; MMP-9 < 1 nM
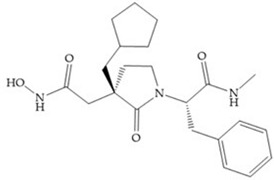 **Ki**: MMP-1 = 1.3 nM; MMP-2 = 1.1 nM; MMP-3 = 187 nM	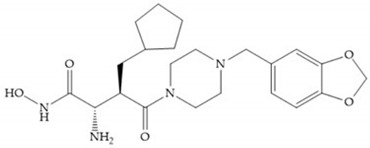 **Ki**: MMP-1 = 6.5 nM; MMP-2 = 20 nM; MMP-3 = 240 nM			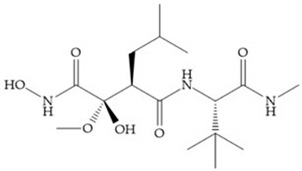 **IC_50_**: MMP-1 = 6 nM; MMP-2 = 30 nM; MMP-3 = 40 nM
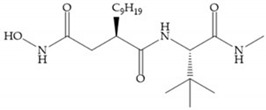 **IC_50_**: MMP-1 = 375 nM; MMP-2 < 0.15 nM; MMP-3 = 18 nM; MMP-9 = 1.5 nM	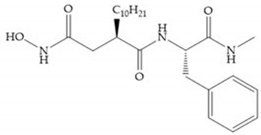 **IC_50_**: MMP-1 = 20 nM; MMP-2 = 2 nM; MMP-3 = 100 nM; MMP-9 = 2 μM			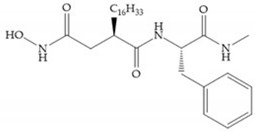 **IC_50_**: MMP-2 = 20 nM; MMP-3 = 300 nM; MMP-9 = 1 nM
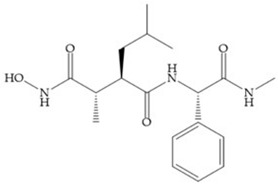 KB-R7785 **IC_50_**: MMP-1 = 3 nM; MMP-2 = 7.5 nM; MMP-3 = 1.9 nM; MMP-9 = 3.9 nM	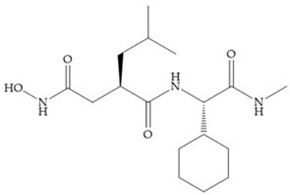 **IC_50_**: MMP-1 = 5.4 nM; MMP-2 = 8.4 nM; MMP-3 = 2.3 nM; MMP-9 = 5 nM; MMP-14 = 2.3 nM			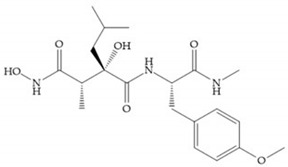 **IC_50_**: MMP-1 = 5 nM; MMP-2 = 1 nM; MMP-3 = 15 nM; MMP-9 = 1 nM
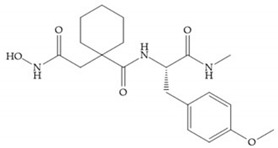 **IC_50_**: MMP-1 = 150 nM	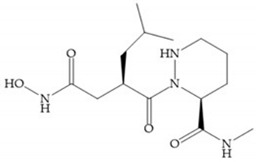 **IC_50_**: MMP-3 = 300 nM			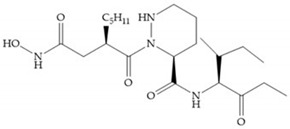 Matlystatin B **IC_50_**: MMP-2 = 1.7 μM; MMP-9 = 570 nM
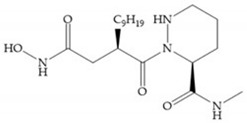 R-94138 **IC_50_**: MMP-2 = 38 nM; MMP-3 = 28 nM; MMP-7 = 23 nM; MMP-9 = 1.2 nM; MMP-13 = 38 nM	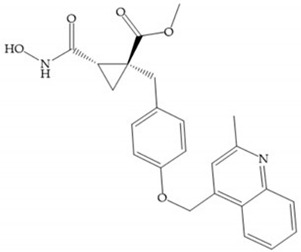 **Ki**: MMP-1 > 8.3 μM; MMP-2 = 3.4 μM; MMP-3 = 1.3 μM; MMP-7 > 8.3 μM; MMP-14 = 7.7 μM			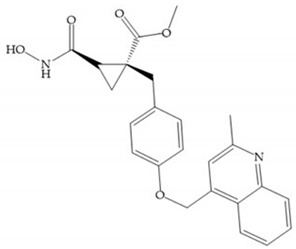 **Ki**: MMP-1 > 12.5 μM; MMP-2 = 1.1 μM; MMP-3 = 100 nM; MMP-7 = 200 nM; MMP-14 = 1.8 nM
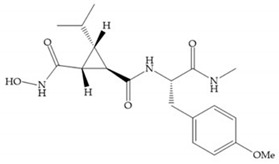 **Ki**: MMP-1 = 200 nM	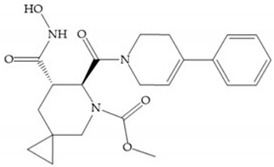 **IC_50_**: MMP-1 > 5 μM; MMP-2 = 35 nM; MMP-3 = 3.56 nM; MMP-7 > 5 μM; MMP-9 = 304 nM; MMP-12 = 17 nM; MMP-14 = 772 nM; MMP-15 = 60 nM			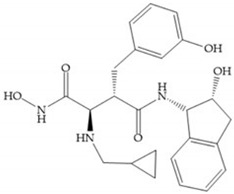 **Ki**: MMP-1 = 33.16 μM; MMP-2 = 6.3 μM; MMP-8 = 171 nM; MMP-9 = 4.468 μM
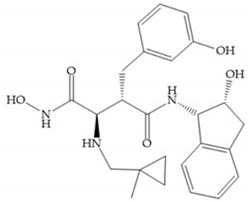 **Ki**: MMP-1 = 5.248 μM; MMP-2 > 3 μM; MMP-3 > 4.5 μM; MMP-9 < 1 nM; MMP-13 > 5 μM	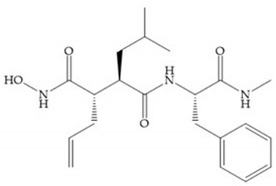 BB-1101 **IC_50_**: MMP-1 = 10 nM; MMP-2 = 5 nM; MMP-3 = 30 nM; MMP-7 = 30 nM; MMP-8 = 3 nM; MMP-9 = 3 nM			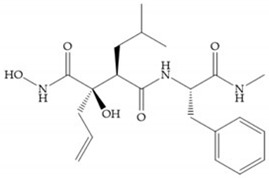 **IC_50_**: MMP-1 = 3.1 nM; MMP-2 = 4.2 nM; MMP-3 = 25 nM
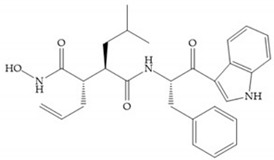 **IC_50_**: MMP-1 = 1.1 nM; MMP-2 = 1.1 nM; MMP-3 = 2.3 nM; MMP-7 = 2.2 nM	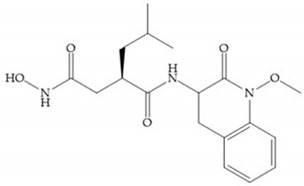 OPB-3206 **IC_50_**: MMP-1 = 700 nM; MMP-2 = 5 μM; MMP-3 = 2 μM; MMP-9 = 500 nM			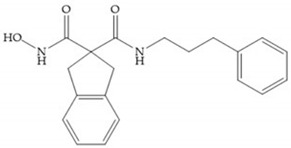 **IC_50_**: MMP-8 = 300 nM
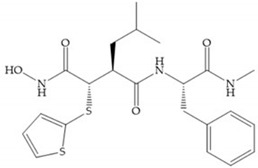 Batimastat (BB-94) **IC_50_**: MMP-1 = 3 nM; MMP-2 = 4 nM; MMP-3 = 20 nM; MMP-7 = 6 nM; MMP-8 = 10 nM; MMP-9 = 1 nM; MMP-13 = 1 nM; MMP-14 = 2.8 nM **Ki**: MMP-1 = 10 nM; MMP-2 = 4 nM; MMP-3 = 20 nM; MMP-8 = 10 nM; MMP-9 = 1 nM	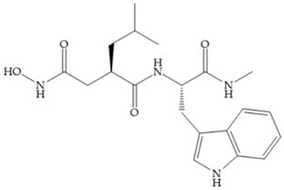 Ilomastat (GM6001; Galardin^®^) **IC_50_**: MMP-1 = 0.4 nM; MMP-2 = 0.4 nM; MMP-3 = 0.19 nM; MMP-14 = 5.2 nM **Ki**: MMP-1 = 0.4 nM; MMP-2 = 0.39 nM; MMP-3 = 26 nM; MMP-8 = 0.18 nM; MMP-9 = 0.2 nM			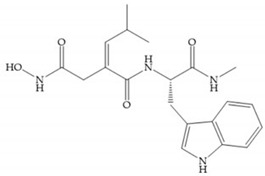 Analogue of Ilomastat **IC_50_**: MMP-2 = 1.3 nM; MMP-3 = 179 nM
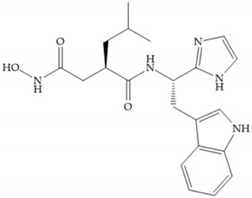 **IC_50_**: MMP-1 = 3 nM; MMP-3 = 280 nM; MMP-7 = 18 nM	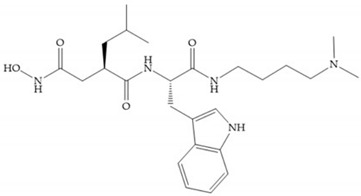 **Ki**: MMP-1 = 3 nM			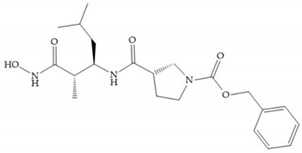 **IC_50_**: MMP-1 = 8 μM; MMP-2 = 8 μM; MMP-3 = 3.5 μM
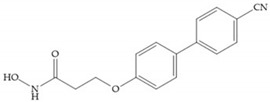 **IC_50_**: MMP-1 = 3.3 μM; MMP-2 = 32 nM; MMP-3 = 57 nM	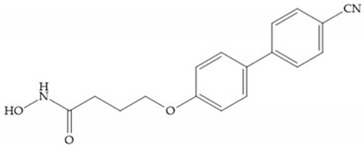 **IC_50_**: MMP-3 = 3.4 μM			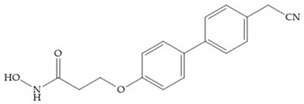 **IC_50_**: MMP-3 = 15 nM
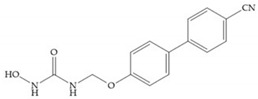 **IC_50_**: MMP-1 > 50 μM; MMP-2 > 120 μM; MMP-3 = 80 μM; MMP-8 > 120 μM	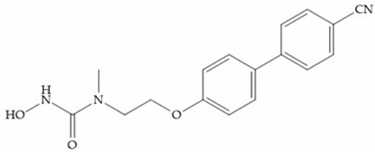 **IC_50_**: MMP-2 = 52 μM; MMP-3 = 200 μM; MMP-8 = 1200 μM			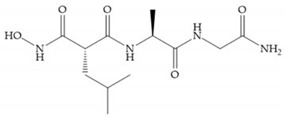 **IC_50_**: MMP-8 = 121 μM
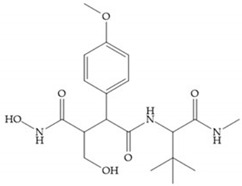 PKF 242-484 **Ki**: MMP-1 = 3.6 nM; MMP-2 = 0.1 nM; MMP-3 = 0.9 nM; MMP-9 = 1 nM; MMP-13 = 4.5 nM	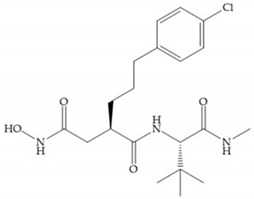 CT1746 **Ki**: MMP-1 = 122 nM; MMP-2 = 0.04 nM; MMP-3 = 10.9 nM; MMP-7 = 136 nM; MMP-9 = 0.17 nM			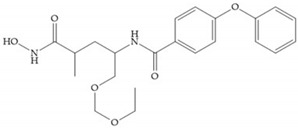 ONO-4817 **IC_50_**: MMP-1 = 1600 nM; MMP-9 = 2.1 nM **Ki**: MMP-2 = 0.73 nM; MMP-3 = 42 nM; MMP-7 = 2500 nM; MMP-12 = 0.45 nM; MMP-13 = 1.1 nM
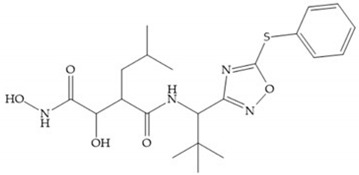 AS 111793# **IC_50_**: MMP-1 = 20 nM			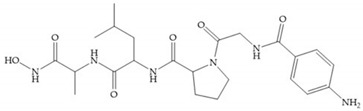 MMPI-I **IC_50_**: MMP-1 = 1 μM; MMP-3 = 150 μM; MMP-8 = 1 μM; MMP-9 = 30 μM	
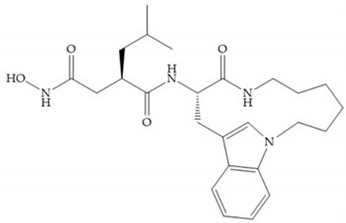 **IC_50_**: MMP-1 = 0.1 nM; MMP-3 = 9 nM; MMP-8 = 0.4 nM; MMP-9 = 0.2 nM		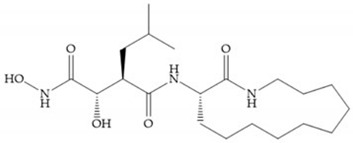 **IC_50_**: MMP-1 = 30 nM; MMP-2 = 20 nM; MMP-3 = 500 nM; MMP-7 = 200 nM; MMP-8 = 20 nM		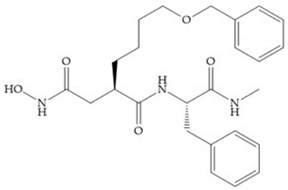 **Ki**: MMP-1 = 1450 nM; MMP-3 = 15 nM; MMP-8 = 2 nM; MMP-9 = 3 nM
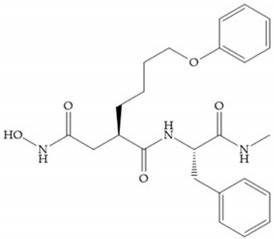 **Ki**: MMP-1 = 8 nM; MMP-3 = 28 nM; MMP-8 < 2 nM; MMP-9 = 1 nM			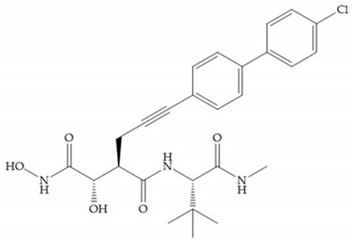 **IC_50_**: MMP-1 = 100 nM; MMP-2 = 0.07 nM; MMP-3 = 3 nM; MMP-7 = 700 nM; MMP-8 = 4 nM; MMP-9 = 1 nM	
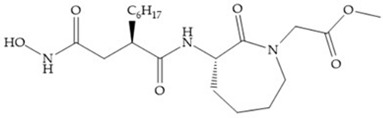 **IC_50_**: MMP-1 = 11 nM; MMP-3 = 1.04 μM		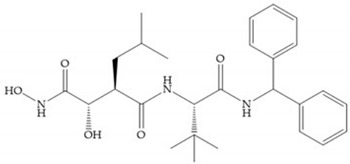 **IC_50_**: MMP-1 = 600 nM; MMP-2 = 3 μM; MMP-3 = 50 nM; MMP-7 = 4 nM		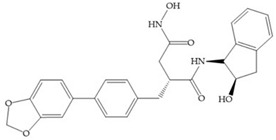 **Ki**: MMP-1 = 7.56 μM; MMP-7 = 622 nM; MMP-13 = 7.3 nM
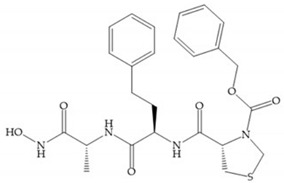 **IC_50_**: MMP-1 = 6 μM; MMP-2 = 200 nM; MMP-3 = 100 nM		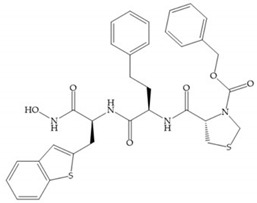 **IC_50_**: MMP-2 = 5 nM		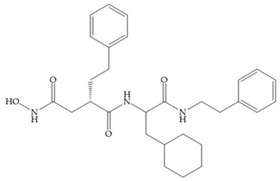 **Ki**: MMP-2 = 2.2 nM
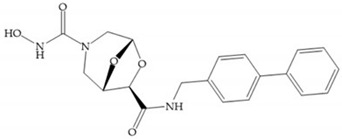 **IC_50_**: MMP-7 = 1 510 μM; MMP-12 = 149 μM		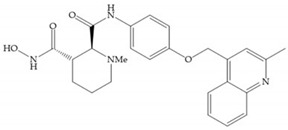 **Ki**: MMP-1 > 4.949 μM; MMP-2 > 3.333 μM; MMP-9 > 2.128 μM		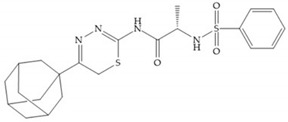 **Ki**: MMP-2 > 15 μM; MMP-8 > 15 μM; MMP-9 > 15 μM; MMP-12 = 410 nM; MMP-13 > 15 μM; MMP-14 = 3.07 μM
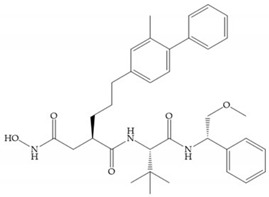 **IC_50_**: MMP-1 = 5.9 μM; MMP-2 = 750 nM; MMP-3=2.1 nM; MMP-9 = 560 nM; MMP-14 = 930 nM		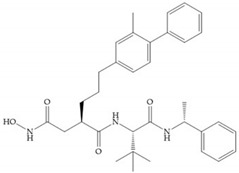 **IC_50_**: MMP-1 = 51 μM; MMP-2 = 1.79 μM; MMP-3 = 5.9 nM; MMP-9 = 840 nM; MMP-13 = 73 nM; MMP-14 = 1.9 μM		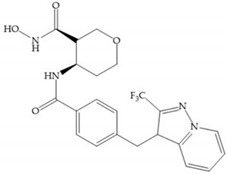 **Ki**: MMP-1 > 4.946 μM; MMP-2 > 3.333 μM; MMP-3 > 4.501 μM; MMP-7 > 6.368 μM; MMP-8 > 3.058 μM; MMP-9 > 2.128 μM; MMP-10 > 5.346 μM; MMP-12 > 6.023 μM; MMP-13 > 5.025 μM; MMP-14 > 5.290 μM; MMP-15 > 7.088 μM
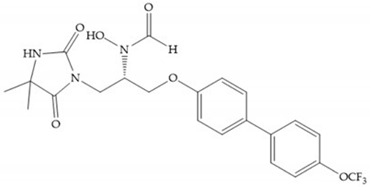 **IC_50_**: MMP-1 = 4.6 μM; MMP-2 = 4 nM; MMP-3 = 42 nM; MMP-7 > 10 μM; MMP-9 = 120 nM		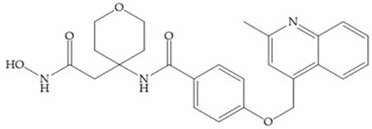 **Ki**: MMP-1 > 4.494 μM; MMP-2 > 3.333 μM; MMP-3 = 82 nM; MMP-7 = 25 nM; MMP- 8 > 3.1 μM; MMP-9 > 2.128 μM; MMP-13 > 5.025 μM; MMP-14 > 5.290 μM; MMP-15 > 7.088 μM; MMP-16 > 5.554 μM		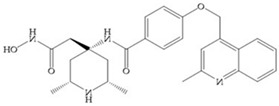 **Ki**: MMP-1 > 5 μM; MMP-2 > 3 μM; MMP-3 = 762 nM; MMP-8 = 2.05 μM; MMP-9 > 3 μM; MMP-10 > 1.650 μM; MMP-13 > 5 μM; MMP-14 = 163 nM; MMP-15 = 1.7 μM
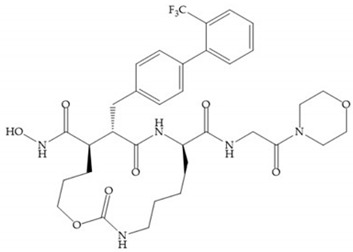 **Ki**: MMP-1 > 2 μM; MMP-2 > 2 μM; MMP-3 > 2 μM; MMP-7 = 834 nM; MMP-8 = 126 nM; MMP-9 > 2 μM; MMP-10 > 2 μM; MMP-12 > 2 μM; MMP-13 = 653 nM; MMP-14 > 2 μM; MMP-15 > 2 μM; MMP-16 > 2 μM			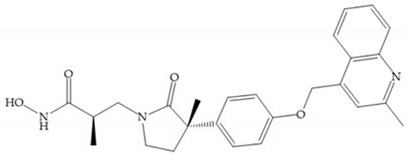 **Ki**: MMP-1 = 30 μM; MMP-2 = 2.05 μM; MMP-3 = 141 nM; MMP-7 = 259 nM; MMP-8 = 257 nM; MMP-9 = 10.34 μM; MMP-13 = 1.417 μM; MMP-14 = 15.872 μM; MMP-15 = 3.997 μM; MMP-16 = 1.599 μM	

**Table 7 biomolecules-10-00717-t007:** IC_50_ and Ki values of sulfonamide hydroxamic acid-based inhibitors.

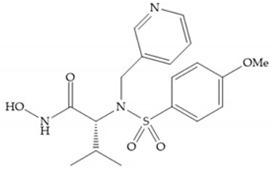 CGS-27023A (MMI270) **IC_50_**: MMP-1 = 33 nM; MMP-2 = 11 nM; MMP-3 = 13 nM; MMP-9 = 8 nM; MMP-12 = 7.7 nM; MMP-13 = 6 nM **Ki****:** MMP-1 = 3 nM; MMP-2 = 20 nM; MMP-3 = 148 nM; MMP-8 = 1.9 nM	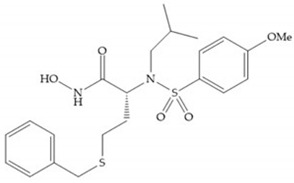 Analogue of CGS-27023A **IC_50_**: MMP-1 = 104 nM; MMP-2 = 0.7 nM; MMP-3 = 0.7 nM; MMP-9 = 2.5 nM; MMP-13 = 12 nM				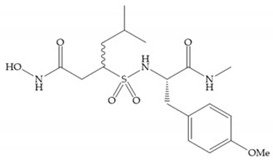 **Ki**: MMP-1 = 770 nM; MMP-2 = 620 nM; MMP-3 = 4.1 μM; MMP-9 = 620 nM
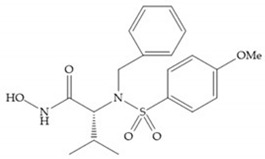 CGS-25966 **Ki**: MMP-3 = 92 nM	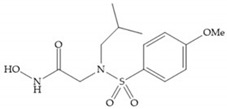 NNGH **IC_50_**: MMP-12 = 72 nM **Ki**: MMP-10 = 0.6 nM	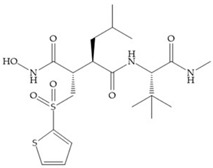 **IC_50_**: MMP-1 = 2 nM; MMP-2 = 20 nM; MMP-3 = 30 nM; MMP-7 = 20 nM; MMP-9 = 7 nM			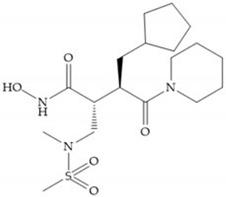 **IC_50_**: MMP-1 = 6 nM; MMP-2 = 900 nM; MMP-3 = 200 M; MMP-7 = 200 nM; MMP-8 = 200 nM; MMP-9 = 2μM; MMP-13 = 400 nM
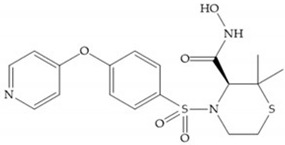 Prinomastat (AG3340) **Ki**: MMP-1 = 8.3 nM; MMP-2 = 0.05 nM; MMP-3 = 0.3 nM; MMP-7 = 54 nM; MMP-9 = 0.26 nM; MMP-13 = 0.03 nM; MMP-14 = 0.33 nM	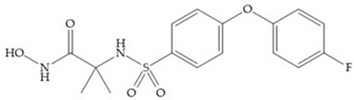 CP-471,474 **IC_50_**: MMP-1 = 1170 nM; MMP-2 = 0.7 nM; MMP-3 = 16 nM; MMP-9 = 13 nM				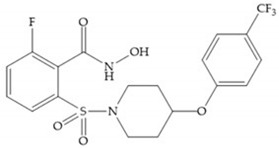 **IC_50_**: MMP-1 > 10 μM; MMP-2 = 3.3 nM; MMP-13 = 12 nM
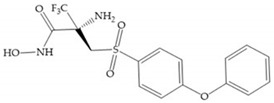 **IC_50_**: MMP-1 = 196 nM; MMP-2 = 0.01 nM; MMP-9 = 1 nM	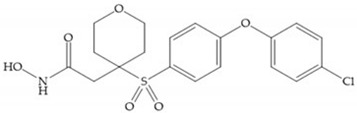 RS-113,456 **IC_50_**: MMP-3 = 5.2 nM **Ki**: MMP-1 = 70 nM; MMP-2 = 0.054 nM; MMP-7 = 240 nM; MMP-8 = 0.13 nM; MMP-9 = 0.065 nM; MMP-12 = 0.15 nM; MMP-13 = 0.17 nM; MMP-14 = 0.089 nM				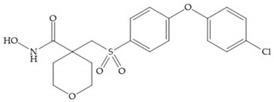 RS-130,830 **Ki**: MMP-1 = 590 nM; MMP-2 = 0.22 nM; MMP-3 = 9.3 nM; MMP-7 = 1.2 μM; MMP-9 = 0.58 nM; MMP-13 = 0.52 nM
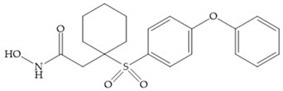 RS-104,966 **Ki**: MMP-1 = 23 nM; MMP-13 = 0.13 nM	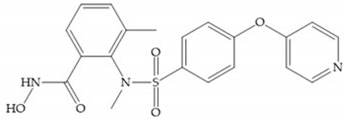 **IC_50_**: MMP-1 = 3 245 nM; MMP-9 = 7 nM; MMP-13 = 4 nM				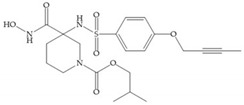 **IC_50_**: MMP-2 = 960 nM; MMP-13 = 1.17 μM; MMP-14 = 3.41 μM
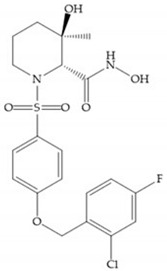 **IC_50_**: MMP-1 = 310 nM	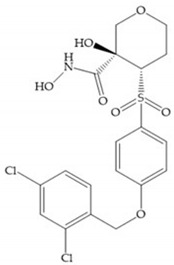 **IC_50_**: MMP-1 = 920 nM; MMP-13 = 0.95 nM	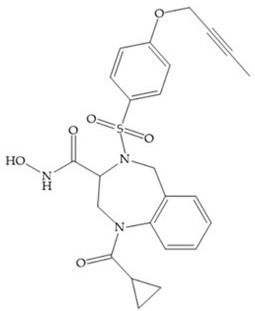 **IC_50_**: MMP-1 = 841 nM; MMP-9 = 33 nM; MMP-13 = 29 nM			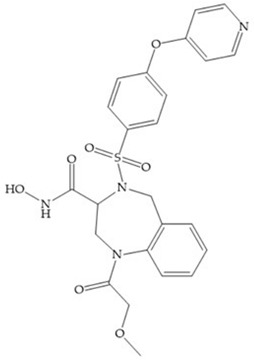 **IC_50_**: MMP-1 = 763 nM; MMP-9 = 2 nM; MMP-13 = 2 nM
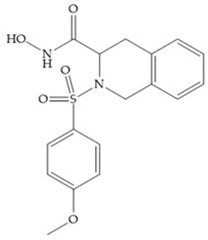 MMP-8 inhibitor I **IC_50_**: MMP-8 = 4 nM	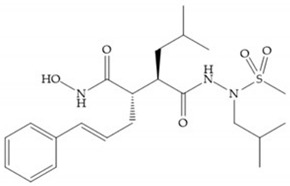 Ro-32-7315 **IC_50_**: MMP-1 = 500 nM; MMP-2 = 250 nM; MMP-3 = 210 nM; MMP-7 = 310 nM; MMP-9 = 100 nM; MMP-12 = 11 nM; MMP-13 = 110 nM			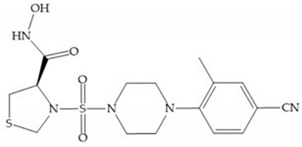 **IC50:** MMP-1 = 346 μM; MMP-9 = 24 μM	
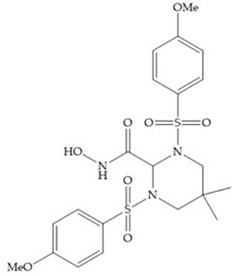 PGE-4410186 **IC_50_**: MMP-1 = 24 nM; MMP-3 = 18.4 nM; MMP-7 = 30 nM; MMP-9 = 2.7 nM	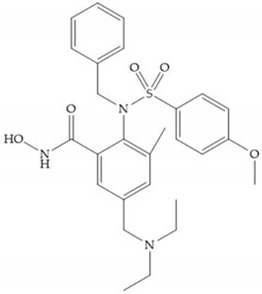 MMP-9 inhibitor I **IC_50_**: MMP-1 = 1.05 nM; MMP-9 = 5 nM; MMP-13 = 113 nM			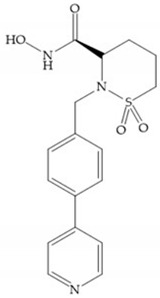 **Ki**: MMP-1 = 1.085 μM; MMP-2 = 1 nM; MMP-9 = 10 nM; MMP-13 = 3 nM	
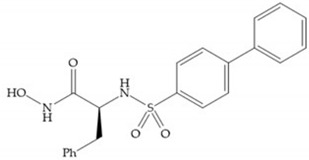 MMPI-II (MMP-2/MMP-9 inhibitor II) **IC_50_**: MMP-1 = 970 nM; MMP-2 = 17 nM; MMP-3 > 1000 nM; MMP-7 = 800 nM; MMP-9 = 30 nM; MMP-14 = 17 nM	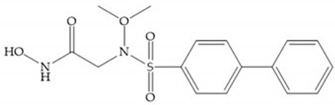 **IC_50_**: MMP-1> 50 μM; MMP-2 = 12 nM; MMP-3 = 4.5 μM; MMP-7 > 50 μM; MMP-9 = 200 nM			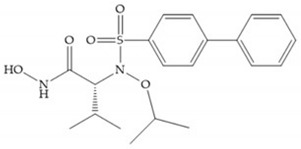 **IC_50_**: MMP-1 = 147 nM; MMP-2 = 0.09 nM; MMP-3 = 50 nM; MMP-7 > 1 μM; MMP-8 = 1.6 nM; MMP-9 = 6.7 nM; MMP-14 = 9.8 nM	
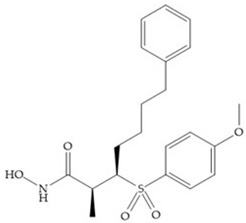 **Ki**: MMP-1 = 2 μM; MMP-2 = 10 nM; MMP-3 = 500 nM	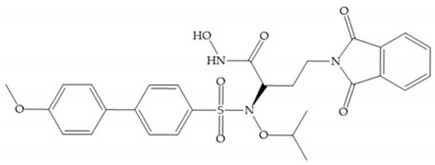 **IC_50_**: MMP-1 = 200 nM; MMP-9 = 0.43 nM			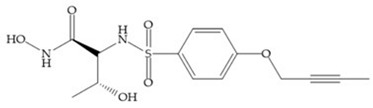 **IC_50_**: MMP-1 = 2.471 μM; MMP-7 = 961 nM; MMP-8 = 35 nM; MMP-9 = 777 nM; MMP-13 = 96 nM; MMP-14 = 582 nM	
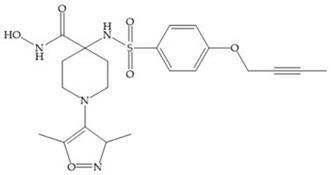 **IC_50_**: MMP-1 = 37.3 μM; MMP-2 = 664 nM; MMP-9 = 5.5 μM; MMP-13 = 2.277 μM; MMP-14 = 24 μM	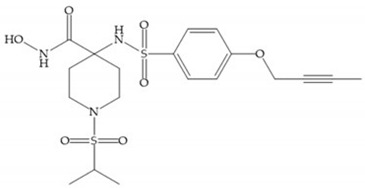 **IC_50_**: MMP-1 = 8.78 μM; MMP-2 = 355 nM; MMP-9 = 1.67 μM; MMP-13 = 230 nM; MMP-14 = 4.71 μM			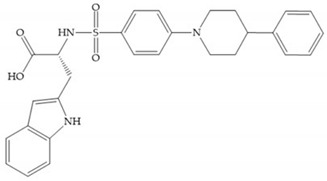 **Ki**: (S enantiomer) MMP-3 = 19 nM (R enantiomer) MMP-3 = 36 nM	
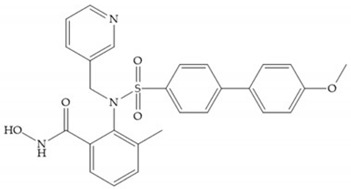 **IC_50_**: MMP-1 = 2.268 μM; MMP-9 = 152 nM; MMP-13 = 18 nM			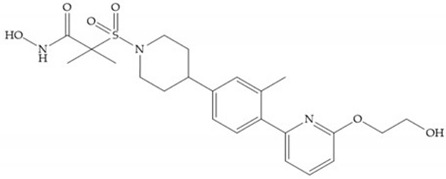 **IC_50_**: MMP-1 = 14 μM; MMP-2 = 529 nM; MMP-3 = 1 nM; MMP-9 = 2.42 μM; MMP-14 = 20.1 μM		

**Table 8 biomolecules-10-00717-t008:** IC_50_ and Ki values of phosphamides hydroxamic acid-based inhibitors.

** 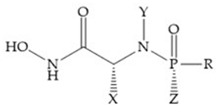 **	IC_50_ (X = H; Y = (CH_2_)_2_C_6_H_5_; Z = Me; R = Ph): MMP-1 = 252 nM; MMP-3 = 700 nM IC_50_ (X = H; Y = (CH_2_)_2_C_6_H_5_; Z = Ph; R = Ph): MMP-1 = 854 nM; MMP-3 = 1.75 μM IC_50_ (X = Me; Y = CH_2_C_6_H_5_; Z = Me; R = Ph): MMP-1 = 120 nM; MMP-3 = 67.9 nM IC_50_ (X = Me; Y = CH_2_C_6_H_5_; Z = Et; R = Ph): MMP-1 = 608 nM; MMP-3 = 700 nM IC_50_ (X = Me; Y = CH_2_C_6_H_5_; Z = Ph; R = Ph): MMP-1 = 6.79 μM; MMP-3 = 10.3 μM IC_50_ (X = CH_2_i-Pr; Y = CH_2_C_6_H_5_; Z = Me; R = Ph): MMP-1 = 20.5 nM; MMP-3 = 24.4 nM IC_50_ (X = CH_2_i-Pr; Y = CH_2_C_6_H_5_; Z = Me; R = Me): MMP-1 = 518 nM; MMP-3 = 1.04 μM
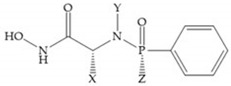	IC_50_ (X = CH_2_i-Pr; Y = H; Z = CH_3_): R isómer, MMP-1 = 2.51 μM; MMP-3 = 2.55 μM S isomer: MMP-1 > 100 μM; MMP-3 = 130.5 μM IC_50_ (X = CH_2_i-Pr; Y = CH_2_C_6_H_5_; Z = CH_3_): R isomer, MMP-1 = 20.5 nM; MMP-3 = 24.4 nM S isomer, MMP-1 = 7.12 μM; MMP-3 = 9.17 μM IC_50_ (X = CH_3_; Y = CH_2_C_6_H_5_; Z = C_2_H_5_): R isomer, MMP-1 = 608 nM; MMP-3 = 700 nM S isomer, MMP-1 = 33.3 μM; MMP-3 = 49.3 μM
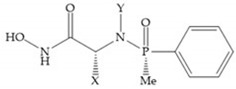	IC_50_ (X = H; Y = (CH_2_)_2_C_6_H_5_): MMP-1 = 525 nM; MMP-3 = 700 nM IC_50_ (X = CH_3_; Y = CH_2_C_6_H_5_): MMP-1 = 120 nM; MMP-3 = 67.9 nM IC_50_ (X = CH_3_; Y = n-C_6_H_13_): MMP-1 = 1.29 μM; MMP-3 = 1.6 μM IC_50_ (X = CH_2_i-Pr; Y = H): MMP-1 = 2.51 μM; MMP-3 = 2.55 μM IC_50_ (X = CH_2_i-Pr; Y = CH_2_C_6_H_5_): MMP-1 = 20.5 nM; MMP-2 = 13.3 nM; MMP-3 = 24.4 nM; MMP-7 = 886 nM; MMP-8 = 5.3 nM; MMP-9 = 20.6 nM; MMP-13 = 7.4 nM

**Table 9 biomolecules-10-00717-t009:** IC_50_ and Ki values of thiolates-based inhibitors.

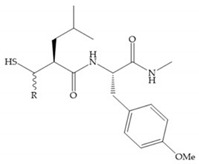 **IC_50_** (R = H): MMP-1 = 360 nM **IC_50_** (R = Me): MMP-1 = 220 nM	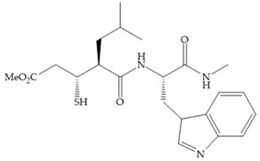 **IC_50_**: MMP-1 = 2.5 nM		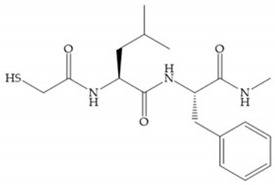 **IC_50_**: MMP-3 = 260 nM; MMP-8 = 50 nM; MMP-9 = 90 nM		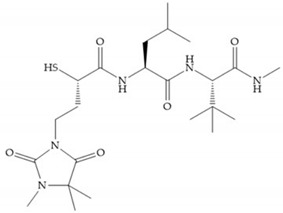 D2163 **IC_50_**: MMP-1 = 25 nM; MMP-2 = 41 nM; MMP-3 = 157 nM; MMP-8 = 10 nM; MMP-9 = 25 nM; MMP-13 = 4 nM
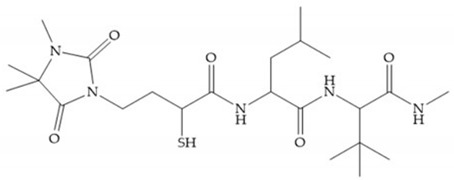 Rebimastat (BMS-275231) **IC_50_**: MMP-1 = 25 nM; MMP-2 = 41 nM; MMP-3 = 157 nM; MMP-9 = 25 nM; MMP-13 = 4 nM			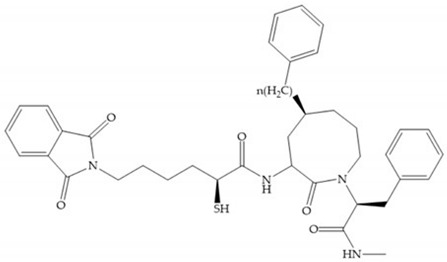 **Ki** (n = 0): MMP-2 = 1.2 nM; MMP-3 = 39 nM; MMP-12 = 18 nM **Ki** (n = 1): MMP-3 = 210 nM		
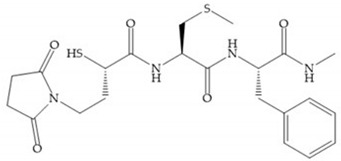 **IC_50_**: MMP-3 = 45 nM; MMP-8 = 3 nM; MMP-9 = 5 nM		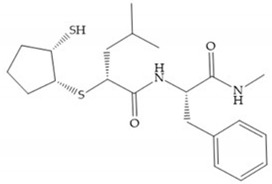 **Ki**: MMP-1 = 49 nM; MMP-2 = 1.1 nM; MMP-3 = 470 nM; MMP-7 = 40 nM; MMP-9 = 0.57 nM; MMP-14 = 24 nM		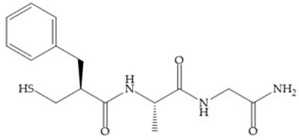 **Ki**: MMP-8 = 1.2 μM	
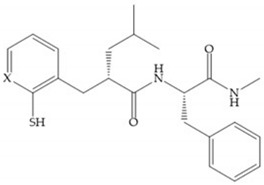 **IC_50_** (X = CH): MMP-1 = 30 nM **IC_50_** (X = N): MMP-1 > 100 μM		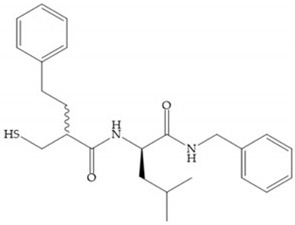 **IC_50_**: MMP-3 = 2.5 μM		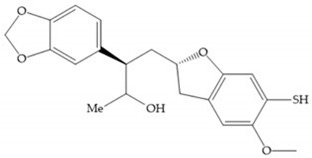 **IC_50_**: MMP-3 = 600 nM	
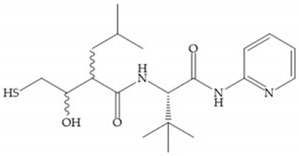 **IC_50_**: MMP-1 = 890 nM; MMP-3 = 4.6 μM; MMP-9 = 4.5 μM		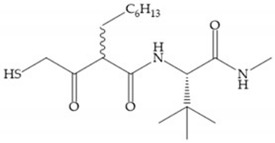 **IC_50_**: MMP-1 = 15 nM; MMP-3 = 16 nM; MMP-9 = 0.3 nM		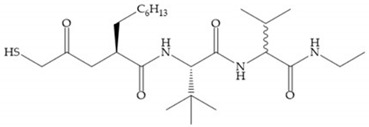 **IC_50_**: MMP-1 > 10 μM; MMP-3 = 36 nM; MMP-9 = 20 nM	
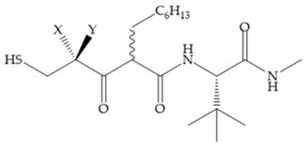 **IC_50_** (X, Y = O): MMP-1 = 10 nM; MMP-2 = 8 nM; MMP-9 = 0.1 nM **IC_50_** (X = OH; Y = H): MMP-1 = 140 nM; MMP-3 = 430 nM; MMP-9 = 12 nM **IC_50_** (X = H; Y = O): MMP-1 = 5 nM; MMP-3 = 9 nM; MMP-9 = 0.14 nM		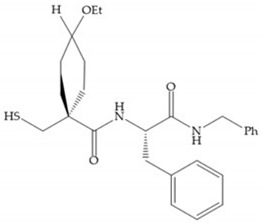 **IC_50_**: MMP-1 = 823 nM; MMP-3 = 207 nM; MMP-9 = 26 nM		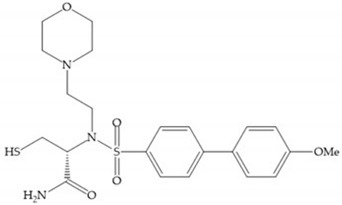 **IC_50_**: MMP-1 = 70 nM; MMP-13 = 0.1 nM	
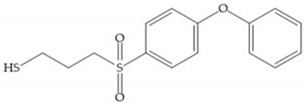 **IC_50_**: MMP-1 = 1.5 μM; MMP-3 = 500 nM; MMP-8 = 4 nM; MMP-13 = 0.5 nM		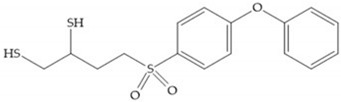 **Ki**: MMP-2 = 46 nM; MMP-3 = 10 μM; MMP-9 = 100 nM; MMP-14 = 210 nM		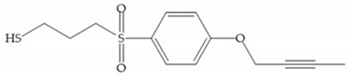 **Ki**: MMP-2 = 1.3 μM; MMP-7 > 2.5 μM; MMP-8 = 2.7 μM; MMP-9 = 6.3 μM; MMP-13 = 1.7 μM	
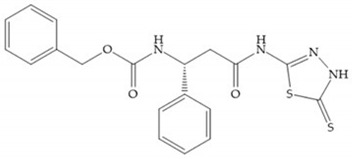 PNU-141803 **Ki**: MMP-2 = 49.5 μM; MMP-3 = 310 nM		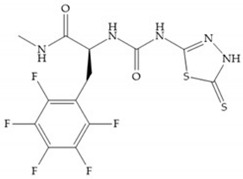 PNU-142372 **Ki**: MMP-2 = 3 μM; MMP-3 = 18 nM		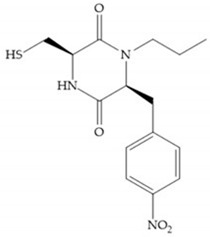 **IC_50_**: MMP-1 = 65 nM; MMP-3 > 20 μM; MMP-9 = 2.9 μM	
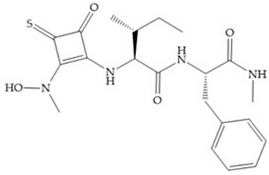 **IC_50_**: MMP-1 = 15 μM		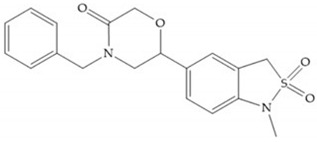 CP-271485 **IC_50_**: MMP-9 = 5.1 μM; MMP-12 > 100 μM		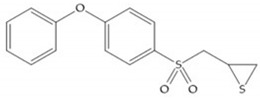 SB-3CT **Ki**: MMP-1 = 206 μM; MMP-2 = 14 nM; MMP-3 = 15 μM; MMP-7 = 96 μM; MMP-9 = 600 nM	
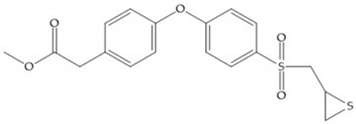 **Ki**: MMP-1 = 11 μM; MMP-2 = 50 nM; MMP-14 = 590 nM		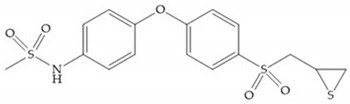 **Ki**: MMP-2 = 16 nM; MMP-3 = 3.6 μM; MMP-7 = 295 μM; MMP-9 = 180 nM; MMP-14 = 900 nM		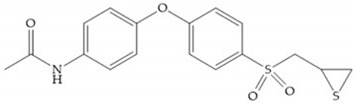 **Ki**: MMP-1 = 5.4 μM; MMP-2 = 110 nM; MMP-3 = 12.2 μM; MMP-7 = 39 μM; MMP-9 = 130 nM; MMP-14 = 680 nM	

**Table 10 biomolecules-10-00717-t010:** IC_50_ and Ki values of carboxylates-based inhibitors.

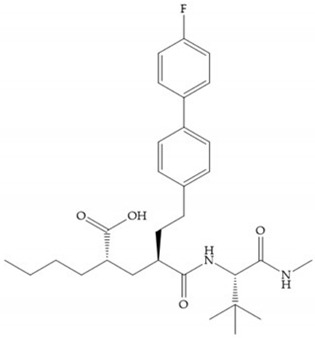 L-758,354 **Ki**: MMP-2 = 17 nM; MMP-3 = 10 nM	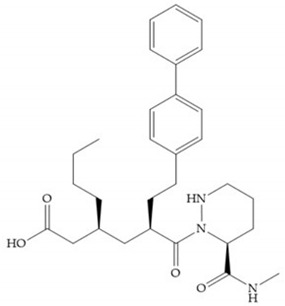 **Ki**: MMP-3 = 42 nM			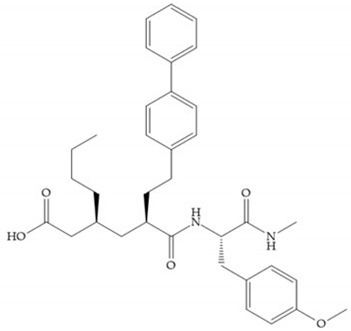 **Ki**: MMP-3 = 21 nM	
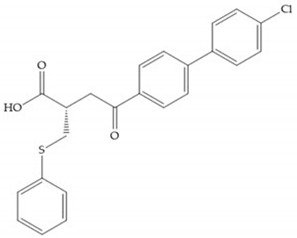 Tanomastat (BAY 12-9566) **Ki**: MMP-1 > 5 μM; MMP-2 = 11 nM; MMP-3 = 134 nM; MMP-9 = 301 nM; MMP-13 = 1.47 μM	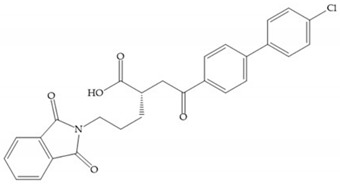 **Ki**: MMP-1 = 0.9 nM; MMP-3 = 15 nM; MMP-9 = 3 nM			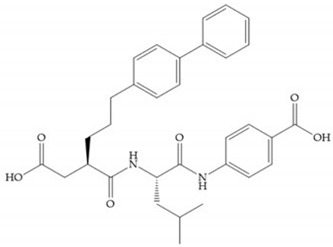 AG 3067 **Ki**: MMP-1 > 1000 nM; MMP-2 = 16 nM; MMP-3 = 2 nM; MMP-7 = 614 nM	
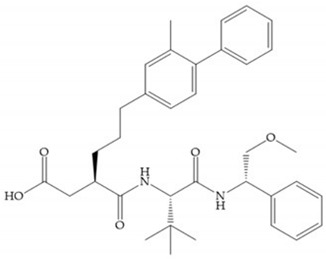 **IC_50_**: MMP-2 = 34.2 μM; MMP-3 = 23 nM; MMP-9 = 30.4 μM; MMP-13 = 2.3 μM; MMP-14 = 66.9 μM	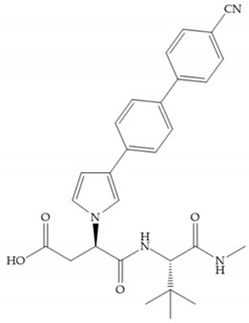 AG 3365 **Ki**: MMP-2 = 0.04 nM; MMP-3 = 1.5 nM; MMP-7 = 305 nM; MMP-13 = 0.05 nM			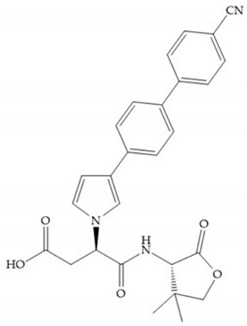 AG 3433 **Ki**: MMP-2 = 0.9 nM; MMP-3 = 19 nM; MMP-7 = 4545 μM; MMP-13 = 3.3 nM	
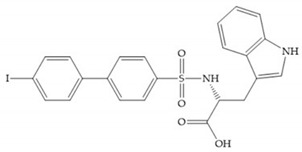 An-1 **IC_50_**: MMP-2 = 9.3 nM; MMP-9 = 201 nM	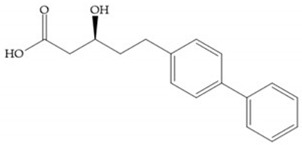 **IC_50_**: MMP-1 > 98 μM; MMP-2 = 4.52 μM; MMP-3 > 98 μM; MMP-7 > 98 μM; MMP-12 = 520 nM; MMP-13 = 12 μM; MMP-14 = 43.5 μM			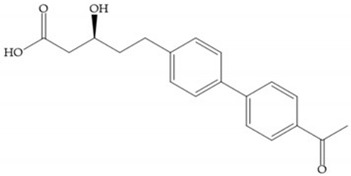 **IC_50_**: MMP-1 > 98 μM; MMP-12 = 62 nM; MMP-13 = 970 nM	
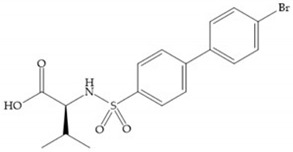 **IC_50_**: MMP-1 = 3.2 μM; MMP-2 = 5 nM; MMP-3 = 12 nM; MMP-9 = 8.3 μM	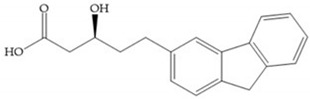 **IC_50_**: MMP-1 > 98 μM; MMP-12 = 1.150 μM; MMP-13 = 26.1 μM			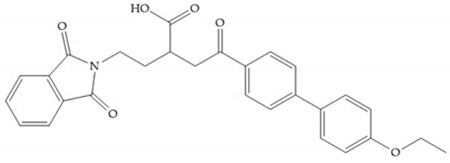 PD-0359601 **IC_50_**: MMP-2 = 6.6 μM; MMP-3 = 3.2 nM; MMP-8 = 160 nM; MMP-12 = 1.7 nM	
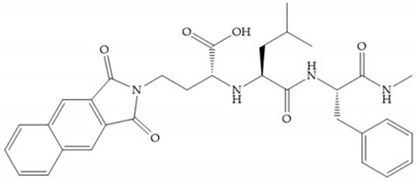 **IC_50_**: MMP-9 = 91 nM **Ki**: MMP-1 = 20 nM; MMP-3 = 91 nM; MMP-9 = 91 nM			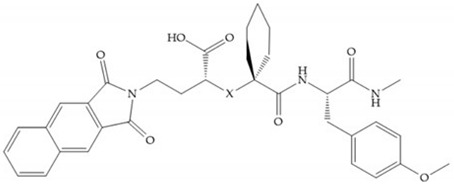 **IC_50_** (X = NH): MMP-1 = 90 nM **IC_50_** (X = CH_2_): MMP-1 = 380 nM		
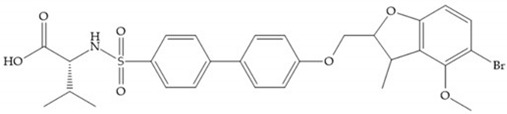 **IC_50_**: MMP-1 > 400 μM; MMP-2 = 132 nM; MMP-3 = 81 nM; MMP-7 = 1.1 μM; MMP-8 = 42 nM; MMP-9 > 7 μM; MMP-13 = 1.8 nM; MMP-14 = 5 μM			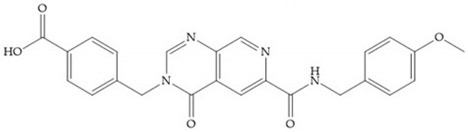 **IC_50_**: MMP-13 = 6.72 nM		
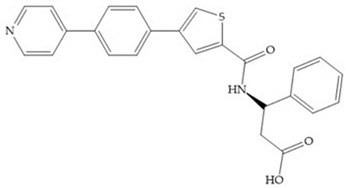 PF-00356231 **IC_50_**: MMP-2 > 100 μM; MMP-3 = 390 nM; MMP-8 = 1.7 μM; MMP-9 = 980 nM; MMP-12 = 14 nM; MMP-13 = 270 nM		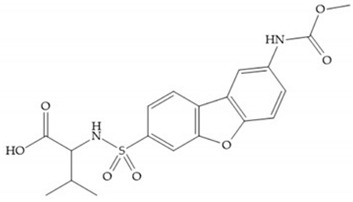 MMP 408 **IC_50_**: MMP-1 > 6 μM; MMP-3 = 351 nM; MMP-7 > 6 μM; MMP-9 = 1.3 μM; MMP-12 = 2 nM; MMP-13 = 120 nM; MMP-14 = 1.1 μM			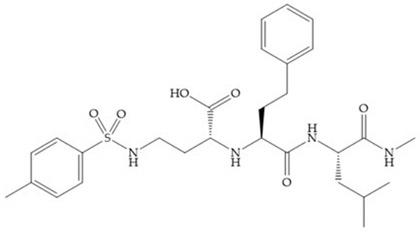 **IC_50_**: MMP-3 = 50 nM
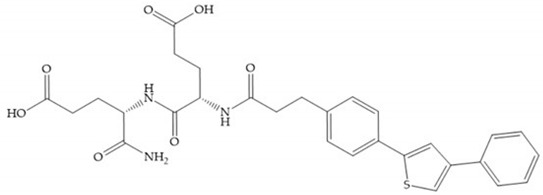 **Ki**: MMP-1 > 10 μM; MMP-2 > 1.06 μM; MMP-3 = 3.88 μM; MMP-7 = 2.01 μM; MMP-8 = 410 nM; MMP-9 > 10 μM; MMP-12 = 1 nM; MMP-13 = 684 nM; MMP-14 = 3.01 μM			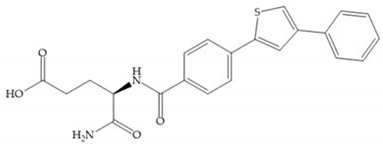 **Ki**: MMP-1 = 127 nM; MMP-3 = 5.819 μM; MMP- = 671 nM; MMP-9 = 2.232 μM; MMP-12 = 2.5 nM; MMP-13 = 501 nM; MMP-14 = 968 nM		
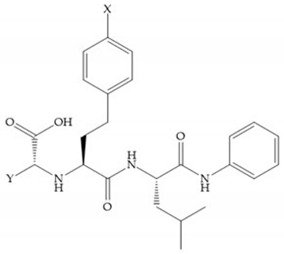 **Ki** (X = H; Y = Me): MMP-1 = 760 nM; MMP-2 = 200 nM; MMP-3 = 470 nM **Ki** (X = C_4_H_9_; Y = Me): MMP-1 = 5.9 μM; MMP-2 = 3.5 nM; MMP-3 = 18 nM **Ki** (X = H; Y = Phthbutyl): MMP-1 = 720 nM; MMP-2 = 86 nM; MM-3 = 8 nM		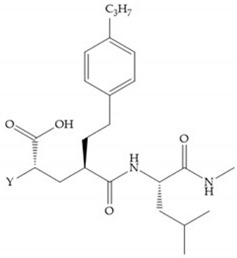 **Ki** (Y = [H2]-Phthbutyl): MMP-1 > 10 μM; MMP-2 = 6 nM; MMP-3 = 0.36 nM **Ki** (Y = Me): MMP-1 > 10 μM; MMP-2 = 310 nM; MMP-3 = 68 nM			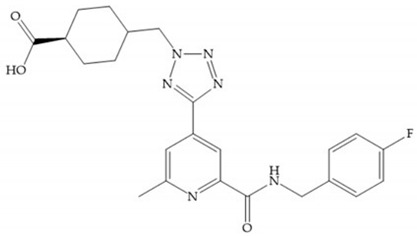 **Ki**: MMP-1 > 25 μM; MMP-2 > 25 μM; MMP-3 > 25 μM; MMP-7 > 25 μM; MMP-8 > 25 μM; MMP-9 > 25 μM; MMP-12 > 25 μM; MMP-13 = 4.4 nM; MMP-14 > 25 μM; MMP-15 > 25 μM; MMP-16 > 25 μM; MMP-24 > 25 μM; MMP-25 > 25 μM; MMP-26 > 25 μM
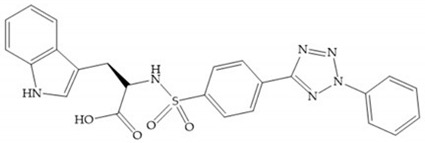 **IC_50_**: MMP-1 > 1000 nM; MMP-2 = 19 nM; MMP-3 > 1000 nM; MMP-7 > 1000 nM; MMP-9 = 32 nM			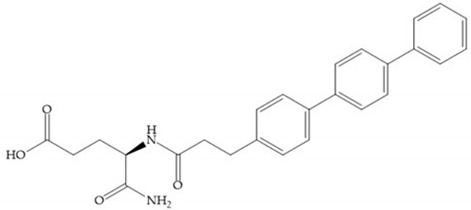 **Ki**: MMP-8 = 205 nM; MMP-12 = 3.3 nM; MMP-13 = 18 nM; MMP-14 = 1.054 μM		
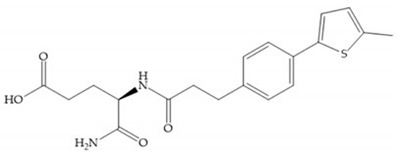 **Ki**: MMP-2 = 57 nM; MMP-3 = 2.164 μM; MMP-8 = 5.3 nM; MMP-13 = 338 nM		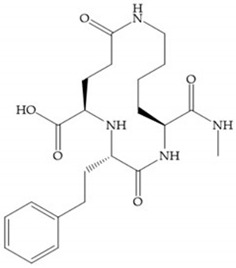 **Ki**: MMP-1 = 2.5 μM; MMP-2 = 8.1 μM; MMP-3 = 13.5 μM; MMP-8 = 17 nM; MMP-9 = 6.6 μM			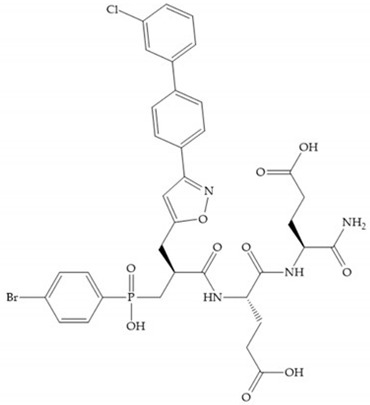 **Ki**: MMP-1 = 67 μM; MMP-2 = 192 nM; MMP-3 = 40 nM; MMP-7 = 626 nM; MMP-8 = 271 nM; MMP-9 = 1.265 μM; MMP-11 = 18.4 μM; MMP-12 = 0.19 nM; MMP-13 = 49 nM; MMP-14 = 140 nM

**Table 11 biomolecules-10-00717-t011:** IC_50_ and Ki values of phosphorus-based inhibitors.

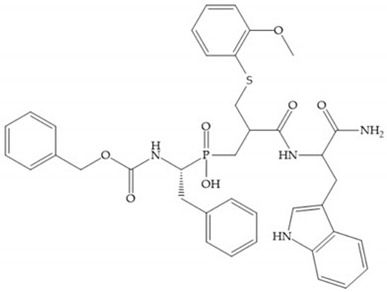 **Ki**: MMP-2 = 30 μM; MMP-8 = 38.9 μM; MMP-9 = 35.6 μM; MMP-11 = 230nM; MMP-13 = 15.7 μM; MMP-14 = 160 μM	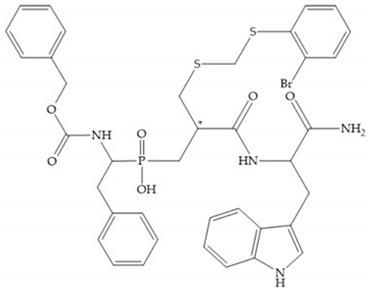 **Ki****:** MMP-2 = 4.65 μM; MMP-8 = 18.4 μM; MMP-9 = 3.91 μM; MMP-11 = 0.11 μM; MMP-13 = 4.7 μM; MMP-14 = 30.1 μM	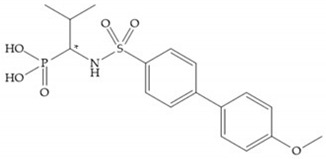 Cyclohexylamine salt of (S/R)-1-(3′-methylbiphenyl-4-sulfonylamino)-methylpropyl phosphonic acid **IC_50_** (S isomer): MMP-3 > 100 μM **IC_50_** (R isomer): MMP-1 = 150 nM; MMP-2 = 1.5 nM; MMP-3 = 52 nM; MMP-7 = 460 nM; MMP-8 = 1.4 nM; MMP-9 = 8 nM; MMP-13 = 2.6 nM; MMP-14 = 79 nM **Ki** (S isomer): MMP-2 = 1.2 μM; MMP-8 = 0.7 μM **Ki** (R isomer): MMP-2 = 5 nM; MMP-3 = 0.04 μM; MMP-8 = 0.6 nM
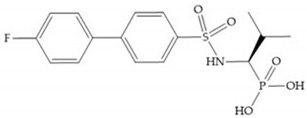 **IC_50_**: MMP-1 = 160 nM; MMP-2 = 20 nM; MMP-3 = 150 nM; MMP-7 = 1.4 μM; MMP-8 = 1.1 nM; MMP-9 = 59 nM; MMP-13 = 13 nM; MMP-14 = 32 nM	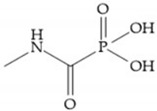 **IC_50_**: MMP-1 > 100 μM; MMP-2 = 0.02 μM; MMP-3 = 90 μM; MMP-8 = 20 μM; MMP-9 > 100 μM	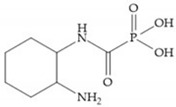 **IC_50_**: MMP-1 > 100 μM; MMP-2 = 4 μM; MMP-3 > 100 μM; MMP-8 > 100 μM; MMP-9 = 20 μM; MMP-12 > 100 μM; MMP-13 > 100 μM
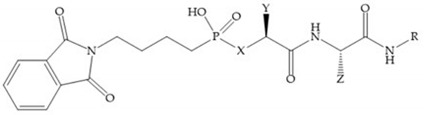 **IC_50_** (X = CH_2_; Y = Ph(CH_2_)_2_; Z = iC_4_H_9_; R = Ph): MMP-1 > 10 μM; MMP-2 = 20 nM; MMP-3 = 1.4 nM **Ki** (X = NH; Y = iC_4_H_9_; Z = 2-naphthyl; R = Me): MMP-3 = 7 nM	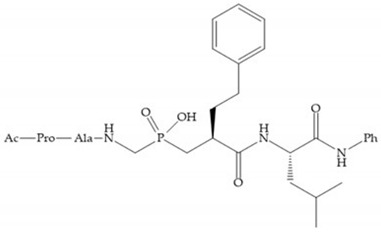 **Ki**: MMP-1 = 3 μM; MMP-3 = 6 nM	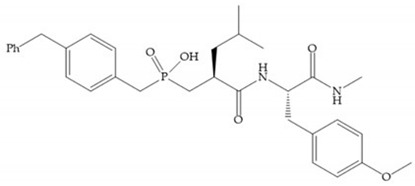 **IC_50_**: MMP-1 = 270 nM
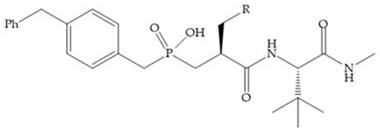 **IC_50_** (R = PhO(CH_2_)_3_): MMP-1 > 30 μM; MMP-13 = 30 nM **IC_50_** (R-PhCH_2_): MMP-1 = 690 nM; MMP-3 = 1.2 μM; MMP-13 = 14 nM	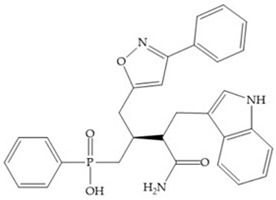 **IC_50_**: MMP-1 > 5 μM; MMP-2 = 1.5 nM; MMP-7 > 2.5 μM; MMP-9 = 13 nM; MMP-13 = 1.6 nM	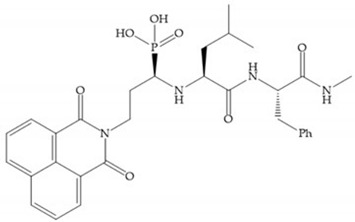 **IC_50_**: MMP-1 = 20 nM
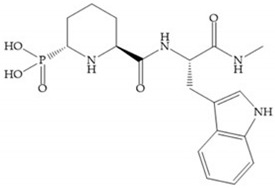 **IC_50_**: MMP-1 > 100 μM; MMP-2 > 100 μM; MMP-9 > 100 μM		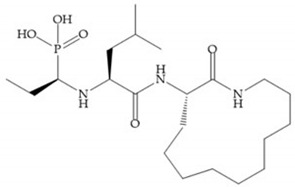 **IC_50_**: MMP-1 = 180 nM

**Table 12 biomolecules-10-00717-t012:** IC_50_ and Ki values of heterocyclic bidentate-based inhibitors.

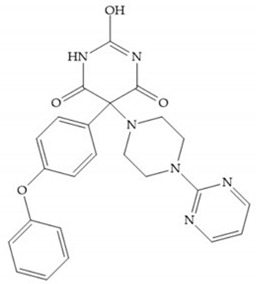 Ro-206-0222 **IC_50_**: MMP-1 = 4.310 μM; MMP-9 = 2 nM	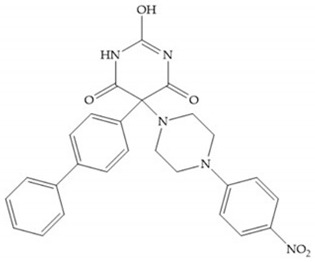 Ro-28-2653 **IC_50_**: MMP-1 = 16 μM; MMP-2 = 12 nM; MMP-3 = 1.8 μM; MMP-8 = 15 nM; MMP-9 = 16 nM; MMP-14 = 10 nM				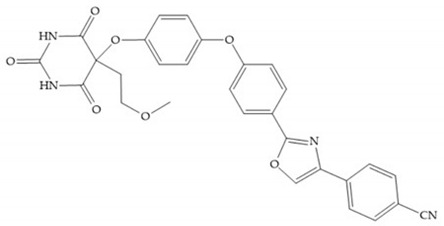 **IC_50_**: MMP-1 = 2.4 μM; MMP-2 = 397 nM; MMP-3 = 17 μM; MMP-8 = 394 nM; MMP-9 = 540 nM; MMP-12 = 619 nM; MMP-13 = 0.36 nM; MMP-14 = 540 nM
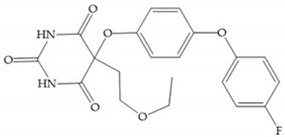 **IC_50_**: MMP-13 = 0.87 nM; MMP-14 = 23 nM	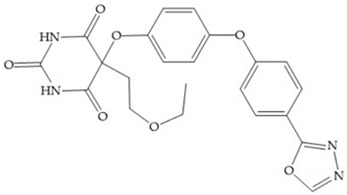 **IC_50_**: MMP-13 = 1 nM; MMP-14 = 220 nM				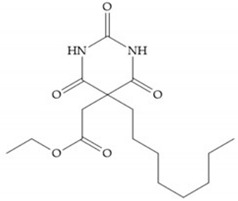 **IC_50_**: MMP-8 = 107 nM; MMP-9 = 20 nM
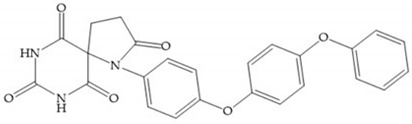 **Ki**: MMP-1 > 5 μM; MMP-2 = 1.8 nM; MMP-9 = 1.9 nM; MMP-13 = 0.33 nM				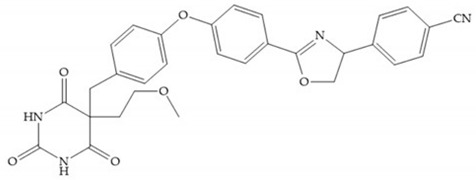 **IC_50_**: MMP-2 = 0.14 μM; MMP-8 = 0.14 μM; MMP-12 = 0.22 μM; MMP-13 = 0.36 nM	
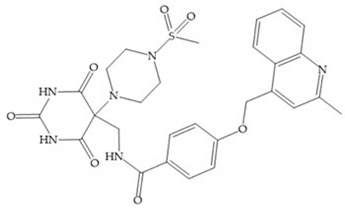 **Ki**: MMP-2 = 2.17 μM; MMP-3 > 4.50 μM; MMP-7 > 6.37 μM; MMP-12 = 1.02 μM		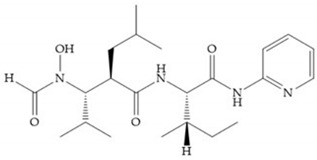 GW-3333 **IC_50_**: MMP-1 = 19 μM; MMP-3 = 20 nM; MMP-9 = 16 nM			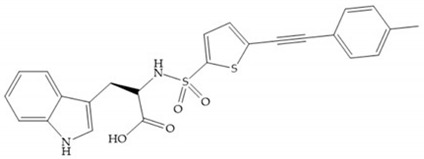 S-3304 **IC_50_**: MMP-2 = 2 nM; MMP-9 = 10 nM
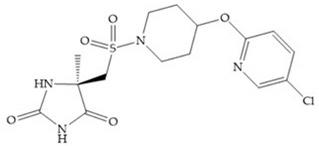 AZD-126 **IC_50_**: MMP-9 = 4.5 nM; MMP-12 = 6.1 nM		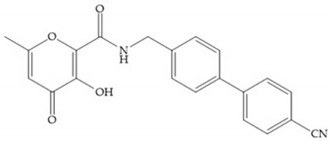 **IC_50_**: MMP-1 > 50 μM; MMP-2 = 610 nM; MMP-3 = 10 nM			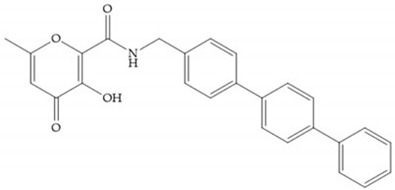 **IC_50_**: MMP-1 > 50 μM; MMP-2 > 50 μM; MMP-3 = 19 nM
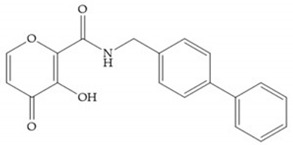 **IC_50_**: MMP-1 > 50 μM; MMP-2 = 4.4 μM; MMP-3 = 77 nM; MMP-7 > 50 μM; MMP-8 = 245 nM; MMP-9 = 32.3 μM; MMP-12 = 85 nM; MMP-13 = 6.6 μM		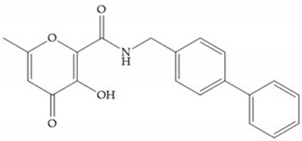 **IC_50_**: MMP-1 > 50 μM; MMP-2 = 9.3 μM; MMP-3 = 0.24 μM; MMP-7 > 50 μM; MMP-8 = 64 nM; MMP-9 > 50 μM; MMP-12 = 22 nM; MMP-13 = 20.6 μM			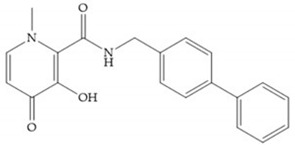 **IC_50_**: MMP-1 > 50 μM; MMP-2 = 16.5 μM; MMP-3 = 41.7 μM; MMP-7 > 50 μM; MMP-8 = 3.8 μM; MMP-9 > 50 μM; MMP-12 = 1.2 μM; MMP-13 = 16.5 μM
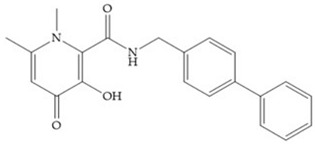 **IC_50_**: MMP-1 > 50 μM; MMP-2 = 7.6 μM; MMP-3> 50 μM; MMP-7 > 50 μM; MMP-8 = 5.0 μM; MMP-9 > 50 μM; MMP-12 = 6.7 μM; MMP-13 = 6.7 μM		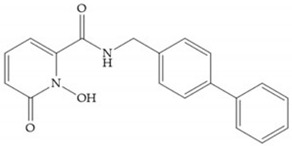 **IC_50_**: MMP-1 > 50 μM; MMP-2 = 0.92 μM; MMP-3 = 0.56 μM; MMP-7 > 50 μM; MMP-8 = 86 nM; MMP-9 = 27.1 μM; MMP-12 = 18 nM; MMP-13 = 4.1 μM			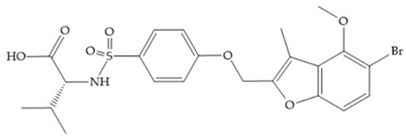 **IC_50_**: MMP-1 > 400 μM; MMP-2 = 135 nM; MMP-3 = 81 nM; MMP-7 = 1.1 μM; MMP-8 = 42 nM; MMP-9 > 7 μM; MMP-13 = 1.8 nM; MMP-14 = 5 μM
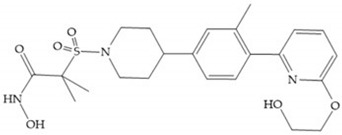 **IC_50_**: MMP-1 = 14 μM; MMP-2 = 529 nM; MMP-3 = 1 nM; MMP-9 = 2.42 μM; MMP-14 = 20.1 μM		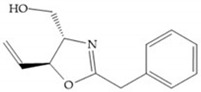 **KI**: MMP-1 > 500 μM; MMP-9 = 6 μM			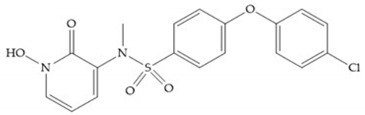 **IC_50_**: MMP-1 > 1 μM; MMP-2 = 5 nM; MMP-3 = 56 nM; MMP-9 = 2.4 nM; MMP-12 = 2.5 nM
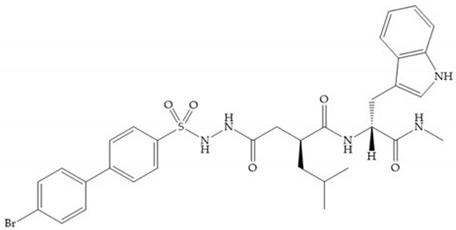 **IC_50_**: MMP-1 = 30 nM; MMP-2 = 9.8 nM; MMP-3 = 1.7 μM; MMP-7 = 475 nM; MMP-9 = 3 nM; MMP-14 = 17 μM		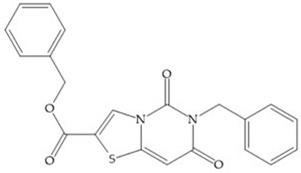 **IC_50_**: MMP-1 > 100 μM; MMP-2 > 100 μM; MMP-3 > 100 μM; MMP-7 > 100 μM; MMP-8 > 100 μM; MMP-9 > 100 μM; MMP-12 > 100 μM; MMP-13 > 100 μM; MMP-14 > 100 μM			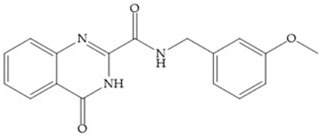 **IC_50_**: MMP-1 > 10 μM; MMP-7 > 10 μM; MMP-9 > 10 μM; MMP-13 = 12 nM; MMP-14 > 10 μM
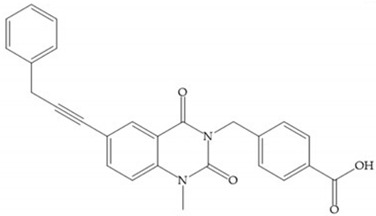 **IC_50_**: MMP-1 > 30 μM; MMP-2 > 30 μM; MMP-3 > 30 μM; MMP-7 > 30 μM; MMP-8 > 100 μM; MMP-9 > 100 μM; MMP-12 > 100 μM; MMP-13 = 0.67 nM; MMP-14 > 30 μM; MMP-17 > 30 μM		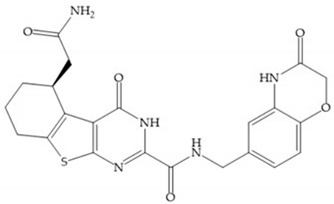 **IC_50_**: MMP-1 > 10 μM; MMP-2 > 10 μM; MMP-3 > 2.5 μM; MMP-7 > 10 μM; MMP-8 = 7.4 nM; MMP-9 > 10 μM; MMP-12 > 10 μM; MMP-14 > 10 μM			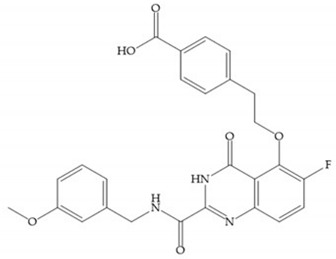 **IC_50_**: MMP-1 > 10 μM; MMP-2 = 5.3 μM; MMP-3 = 4 μM; MMP-7 > 10 μM; MMP-8 = 720 nM; MMP-9 = 10 μM; MMP-10 = 160 nM; MMP-13 = 0.0039 nM; MMP-14 > 10 μM
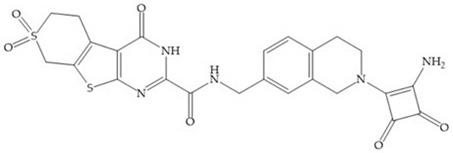 **IC_50_**: MMP-1 > 10 μM; MMP-2 > 2.5 μM; MMP-7 > 10 μM; MMP-8 = 57 nM; MMP-9 > 10 μM; MMP-14 > 10 μM		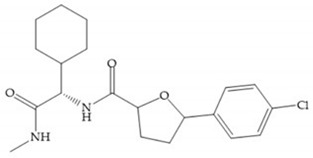 **IC_50_**: MMP-1 > 21.5 μM; MMP-7 > 21.5 μM; MMP-13 = 430 nM			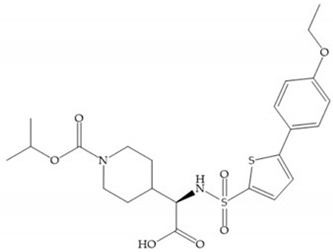 **IC_50_**: MMP-1 > 10 μM; MMP-7 = 3.025 nM; MMP-13 = 0.5 nM
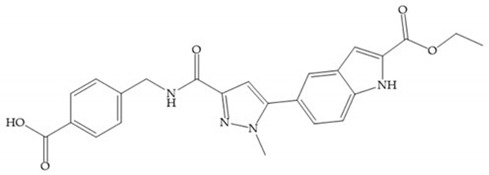 **IC_50_**: MMP-1 > 22 μM; MMP-2 = 18 μM; MMP-3 > 22 μM; MMP-7 > 22μM; MMP-8 > 22 μM; MMP-9 = 8.9 μM; MMP-10 = 16 μM; MMP-12 > 22 μM; MMP-13 = 1 nM; MMP-14 = 8.3 μM			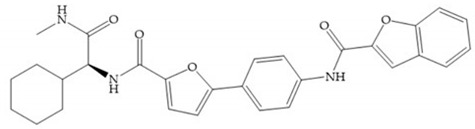 **IC50**: MMP-1 > 18.6 μM; MMP-7 > 18.6 μM; MMP-13 = 620 nM; MMP-14 > 62 μM		
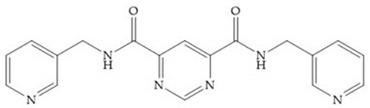 **IC_50_**: MMP-13 = 6.6 μM	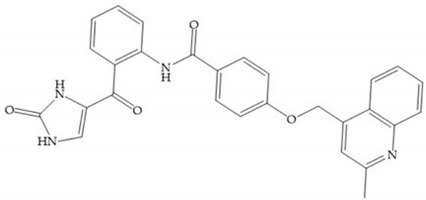 **Ki**: MMP-2 > 3.333 μM; MMP-3 > 4.501 μM; MMP-7 > 636 nM; MMP-12 > 6.023 μM; MMP-13 = 4.314 μM				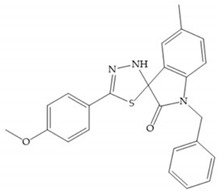 **IC_50_**: MMP-12 > 22 μM; MMP-13 > 100 μM

**Table 13 biomolecules-10-00717-t013:** IC_50_ values of tetracyclines-based inhibitors.

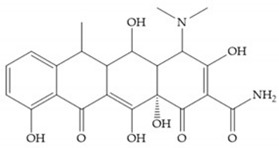 Doxycycline	**IC_50_**: MMP-1 > 400 μM; MMP-7 = 28 μM; MMP-3 = 30 μM; MMP-13 = 2 μM
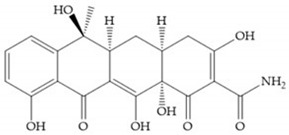 CMT-1	**IC_50_**: MMP-8 = 30 μM; MMP-13 = 1 μM
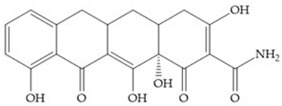 Matastat (COL-3; CMT-3)	**IC_50_**: MMP-1 = 34 μg·mL^−1^; MMP-8 = 48 μg·mL^−1^; MMP-13 = 0.3 μg·mL^−1^
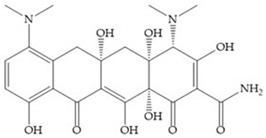 Minocycline	**IC_50_**: MMP-9 = 272 μM

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
