# Peer review of "Challenges in Matrix Metalloproteinases Inhibition"

_biomolecules, 2020, doi:10.3390/biom10050717_

Round 1

Reviewer 1 Report

The important role of MMPs in human biology and pathology is appreciated since about half a century. Due to its important role in diseases such as cancer, rheumatoid arthritis and others the search for inhibitors of MMPs has been long, intense and largely frustrating since almost none of the numerous developed inhibitors made it to the clinic due to severe side effects and other problems.

The review by Laronha et al. is an extensive and very competent description of these developments over the past 30 years.

Major points:

The authors obviously love chemical structures since more than 200 (!) are included in this review. I have to point out that I am not able to verify all these complicated formulas and I hope the authors did this carefully.

I also wonder how important all those formulas are for the general reader of this review.

In detail: Table 6 has 87 formulas, Table 7: 37, Table 9: 30, Table 10: 32, Table 11: 14, Table 12: 37. Many of the compounds shown don’t even have a name. Although I believe that it is part of the value of this review to include all those formulas I would suggest to remove them from the main text and transfer them to an appendix.

One point that I am missing is a discussion of the regulation of MMPs i.e. by inflammatory cytokines. This is an important issue since cytokine inhibitors have been successfully developed and made it to the clinic. However, this could be a different review.

Minor points:

  1. Starting from page 10, the numbering of the figures went wrong. Thus what is figure 3 on this page should be figure 11 and so on. Later we even have figures with 3 digits.
  2. line 16: due to
  3. line 27/28: However excessive degradation of ECM leads to metabolic and immune disease..I think it is the other way around: Metabolic and immune diseases are characterized or leading to excessive degradation of ECM.
  4. Table 1 and 2: some classes are singular some plural (e.g. Collagenases vs Gelatinase)
  5. line 53/54 see point 3
  6. Table 3: TIMP-1 Nucleus of fibroblasts??? TIMP-1 is not expressed in the nucleus but secreted and its expression is regulated by inflammatory cytokines such as IL-6 and TGFß.
  7. lines 72-77, several grammatical errors, please rephrase
  8. lines 93-95, incomplete sentence
  9. line 104: into the cell
  10. lines 122/123 the authors should explain what ADAMs and ADAMTS are.
  11. lines 149. Not the general structure of hydroamic acid is shown but that of hydroxamic acid-based inhibitors
  12. lines 231. The inhibitors do not have an identical structure to collagen!!
  13. line 266. What does NNGH stand for?
  14. line 460. the nitrogen based group is known to the FDA???
  15. line 490-492: several grammatical errors, please rephrase
  16. line 615: side effects of what? The whole paragraph should be rewritten and given an appropriate introduction.

Author Response

Dear Reviewer 1
We in fact intended to list all possible chemical structures since it gives a landscape view of all that has been tried.  The ensemble of selected structures AND their inhibitory properties can be feeded to QSAR programs or additionally  provide inspiration for synthetic chemists.  We didn't include names of the compounds when they were not included in the original publications, but through the structures it is easy to obtain the IUPAC name using common softwares to draw structures like chemdraw.   All the minor points have been carefully addressed and necessary corrections and replacements have been made 1 figures renumbered 2 corrected 3 corrected text according to suggestion 4 corrected to   uniform use of  plural for enzymes 5 corrected  line 53/54 in accordance to point 3  6  Table 3: TIMP-1  expression corrected 7  rephrased lines 72-77,  8  rephrased lines 93-95 9 corrected 10 definition of ADAM/T included 11 claryfied text included  12 lines 231.corrected  13 NNGH definition included 14 line 460. corrected   15 line 490-492:  rephrased 16 line 615:  paragraph rewritten    we appreciate the important contribution to this manuscript   best regards   Jorge Caldeira

Reviewer 2 Report

In the manuscript entitled „ Challenges in matrix metalloproteinases inhibition“ Helena Laronha and colleagues provide a comprehensive overview on synthetic inhibitors of matrix metalloproteinases.

This overview appears very useful. Its publication could be helpful for many scientists who do research in biochemistry, cell biology, pharmacology or medicine.

Still a few improvements appear necessary prior to publication:

  1. The English language needs considerable improvement.
  2. At some points in the text, the subject should be described more precisely.

For example see

--lines 70-80

-- line 88   “controversial”

--line 94

--lines 107-109

line 171 “R-alpha position”

lines 205, 212, 273, ff…. - please check the numbering of the figures

line 365   “4.2.1” – please check section numbering

line 779 ff: check refs 48, 51, 52

Author Response

Dear Reviewer 2

  Thanks for the comments and corrections. We have reviewed the text in respect to language and we hope it is now improved. We took particular care with the specific point that you mentioned and made the necessary corrections   Thanks for the important points addressed.     best regards   Jorge Caldeira  

Reviewer 3 Report

In this manuscript, Laronha et al, aim to outline the current spectrum of MMP inhibitors, with a main focus on synthetic inhibitors. This is a challenging task, given the many existing MMP inhibitors, and is therefore very much appreciated. In addition, many reviews already exist on this topic. Nevertheless, I beleive that the paper has merit, but still requires some important adaptations. Please find below my detailed comments;

  • It is clear that the focus is on synthetic inhibitors while the other inhibitors are explored in less detail. Therefore it might be better to put this focus already in the title. e.g.; Challenges in the inhibition of matrix metallopteinases by synthetic inhibitors. In addition, I am not sure if the title in its current form needs an 's' after 'metalloproteinase'.
  • The authors write that MMPs are enzymes that degrade the extracellular matrix, in particular collagen. This focus is kept in most parts of the review. However, this is a really 'old' view on MMP function. In recent years, the view that MMPs are mainly cleaving extracelluar substrates, being Extracellular matrix, has been put into question. MMPs have been found intracellularly, have intracellular functions and intracellular substrates. also, cell surface molecules, including receptors have been found that are modulated by MMPs. MMPs can cleave and modulate chemokines. Some examples of this can be found in the following papers;PMID: 20812779, PMID: 17562450, PMID: 28526562Collagen and ECM-molecules are not the main substrates for all MMPs! It is very important to be more specific about MMP substrates and functions, since a lack of good knowledge on MMPs is also one of the reasons that MMP inhibitors have/are failing. Please be more specific, this is important.
  • The general structure for MMPs, introduced in figure 4, is a model showing the catalytic domain, linker, hemopexin domain and prodomain. This is a very symplistic model. MMPs also have other domains. I understand that the authors want to use a simple model to later introduce the inhibitor types, however, I wonder why they did decided to include the hemopexin domain. This domain is not present on all MMPs (e.g. MMP-7, MMP-26, MMP-23). Is there a way to change the model?
  • A second comment about the model; The authors show the MMP as a catalytic domain and hemopexin domain (which are almost touching each other) and the prodomain quenched inbetween. In realy this is probably very different since these are most often two terminal domains. Why did the authors decide to display the model in this way?
  • TIMPs are not particularly specific for MMPs. They also inhibit other proteases of the Metzincin clan. Please adjust this. Please see; PMID: 22078297 
  • Besides the menitioned inhibitors, there are also peptide-based inhibitors which might be worth mentioning. These include peptide inhibitors discovered based on substrate and propeptide domain based screens (see PMID: 15801745: PMID: 17017880) or even based on triple helical collage (PMID: 28394608)
  • Table 3; the inhibition mode of MMP-9 by TIMP-1 happens through interaction at two sites; binding to the PEX domain and binding the active site once the prodomain is removed. In addition, inhibition of other Metzincins should be included.
  • Page 4, line 77; "the c-terminal domain allows TIMP and pro-gelatinasses interaction". Please be more specific. There are two interaction sites present (as mentioned above). This might need some further researching.
  • Page 4, the authors state that mechanism of the endogenous inhibitors are unknown. This statement is too strong. the mechanism of action of alpha-2-macroglobulin (a known general protease inhibitor) is well known. In addition, some MMP proteoforms are able to escape regulation by alpha-2-macroglobulin, which opens op potential for selective inhibition. See following references;PMID: 9344465,PMID: 31642940)
  • Some groups have also worked on 'twin' inhibitors, to increase inhibitor selectivity to specific MMP proteoforms. This might be worth mentioning; for example; PMID: 28337319 
  • Another tetracycline which might be usefull to add is minocycline. Another interesting fact is that some antibiotics also reduce expression of MMPs (e.g. Azithromycin). see PMID: 28369077
  • Conclusion; the authors nicely state all the reasons for failure of MMP inhibitors. There might be some other reasons to add;1. It has recently been discovered that several MMP KO mice have problems, including passenger mutations, which might affect previous results with mouse models. see PMID: 261633702. there is still a general lack of knowledge on MMPs - biochemical & biological. (this is illustrate by my second comment)

Author Response

Dear reviewer 3

Thanks for your detailed and important comments corrections and suggestions.

We have addressed now your questions, in the current form of the manuscript. In particular we have included the many references suggested, and clarified the text regarding substrate specificity.

The role of chemokines as MMP substrates have been highlighted, and references were added as suggested.

Concerning our 3D model we have included the hemopexin domain since it is present in majority of the structures, but a remark was added stating that it is absent in some MMPs.

The model is an approximation that serves to have a general view of three-dimensional arrangement of the majority of structures, without de complexity of the atomistic view that is more accurate but less universal.

We build a spatial organization to be representative of the many PDB structures available, to give readers at a glance an informative display of MMPs architecture. However we have now included a sentence regarding the limits of application of the shematic model.

Regarding TIMPS specificity a reference was added as suggested.

In respect to other types of inhibitors (peptides and triple helical collage) references were included now in the manuscript.

Table 3 includes now more details concerning expression and binding.

The TIMP and pro-gelatinases possible interactions were corrected.

The mechanism of the endogenous inhibitors have been clarified and a Reference have been added .

The “Twin inibihtors” reference included

Minocycline is now included and a reference have been added

Other changes were made to address the overall remarks and sugestions corrrections by other reviewers that we think substantial improved the manuscript

With best regards

Jorge Caldeira

Round 2

Reviewer 3 Report

This reviewer greally appreciates the additional efforts done by the authors. They have now addressed all my concerns. I only found some minor errors in the revised version:

Page 2, line 46: typing errors: cytokines, chemokines, growth factors